# Genome-wide association study in 79,366 European-ancestry individuals informs the genetic architecture of 25-hydroxyvitamin D levels

Xia Jiang et al.[#]

Vitamin D is a steroid hormone precursor that is associated with a range of human traits and diseases. Previous GWAS of serum 25-hydroxyvitamin D concentrations have identified four genome-wide significant loci (*GC, NADSYN1/DHCR7, CYP2R1, CYP24A1*). In this study, we expand the previous SUNLIGHT Consortium GWAS discovery sample size from 16,125 to 79,366 (all European descent). This larger GWAS yields two additional loci harboring genome-wide significant variants ($P = 4.7 \times 10^{-9}$ at rs8018720 in *SEC23A*, and $P = 1.9 \times 10^{-14}$ at rs10745742 in *AMDHD1*). The overall estimate of heritability of 25-hydroxyvitamin D serum concentrations attributable to GWAS common SNPs is 7.5%, with statistically significant loci explaining 38% of this total. Further investigation identifies signal enrichment in immune and hematopoietic tissues, and clustering with autoimmune diseases in cell-type-specific analysis. Larger studies are required to identify additional common SNPs, and to explore the role of rare or structural variants and gene–gene interactions in the heritability of circulating 25-hydroxyvitamin D levels.

#A full list of authors and their affliations appears at the end of the paper

Vitamin D is an essential fat soluble vitamin and steroid pro-hormone that plays an important role in musculoskeletal health. Vitamin D deficiency has been linked to autoimmune[1,2] and infectious disease[3], cardiovascular disease[4], cancer[5], and neurodegenerative conditions[6]. Serum 25-hydroxyvitamin D, a primary circulating form of vitamin D and a measure that best reflects vitamin D stores, is influenced by many factors including sun exposure, age, body mass index[7], dietary intake of certain foods such as fortified dairy products and oily fish, supplements, and genetic factors[8]. The concentration of 25-hydroxyvitamin D has been reported to be highly heritable, with heritability estimates of 50–80% from classical twin studies[9,10].

A genome-wide association study (GWAS) meta-analysis of serum 25-hydroxyvitamin D[11] in 4501 participants of European ancestry and replication in 2221 samples identified variants in three loci (group component (*GC*), 7-dehydrochlesterol reductase (*NADSYN1/DHCR7*), and 25-hydroxylase (*CYP2R1*)). A larger GWAS conducted by the SUNLIGHT consortium in 16,125 European ancestry individuals, with a replication sample of 17,871, replicated these three loci and discovered one additional locus (*CYP24A1*)[8]. However, despite these loci being in or near genes encoding proteins involved in vitamin D synthesis, the associated variants collectively explain only a small fraction of the variance in 25-hydroxyvitamin D concentrations (~5%)[8,11,12]. Therefore, to extend our previous findings and better understand the genetic architecture underlying serum 25-hydroxyvitamin D, as well as test for interactions between dietary vitamin D intake and genetic factors, we conducted a large-scale GWAS meta-analysis on this important vitamin.

Our GWAS with a 79,366 discovery sample and a 40,562 replication sample replicates four previous loci and identifies two new genetic loci for serum levels of 25-hydroxyvitamin D. We further find evidence for a shared genetic basis between circulating 25-hydroxyvitamin D and autoimmune diseases. Our analyses suggest a relatively modest SNP-heritability rate of 25-hydroxyvitamin D when considering only common variants. Larger studies are required to identify additional common SNPs, and to explore the role of rare or structural variants. The genetic instruments identified by our results could be used in future Mendelian Randomization analyses of the association between vitamin D and complex traits.

## Results

**Study description and GWAS.** This study represents an expansion of our previous SUNLIGHT consortium GWAS[8]. Here, we combine the 5 discovery cohorts and 5 in-silico replication cohorts from that study, and augment these with 21 additional cohorts that have joined the SUNLIGHT consortium since 2010 (study characteristics are described in Supplementary Table 1, Supplementary Note 1). In contrast to the previous meta-analysis which involved discovery, in-silico and de-novo genotyping stages, we performed a first stage discovery meta-analysis on a total of up to 79,366 individuals and replicated novel findings in two independent separate in-silico data sets (40,562 individuals collected by EPIC and 2195 individuals collected by SOCCS). To assess and control for population stratification, we examined QQ-plots and genomic control inflation factors for each contributing cohort prior to meta-analysis. We did not observe evidence for widespread inflation (median $\lambda_{GC} = 0.92$; only 1/31 samples with $\lambda_{GC} > 1.01$), indicating that our GWAS results were not inflated by population stratification or cryptic relatedness (Supplementary Fig. 1). Despite the slightly deflated $\lambda_{GC}$ observed in some of the constituent cohorts most probably due to over-correction of test statistics, our $\lambda_{GC}$ of 0.99 in all samples indicated appropriate

control for population stratifications and confounders. We identified six susceptibility loci harboring genome-wide significant SNPs, confirming four previously reported loci at *GC* ($P = 4.7 \times 10^{-343}$ at rs3755967), *NADSYN1/DHCR7* ($P = 3.8 \times 10^{-62}$ at rs12785878), *CYP2R1* ($P = 2.1 \times 10^{-46}$ at rs10741657), *CYP24A1* ($P = 8.1 \times 10^{-23}$ at rs17216707), and two novel loci at *AMDHD1* ($P = 1.9 \times 10^{-14}$ at rs10745742) and *SEC23A* ($P = 4.7 \times 10^{-9}$ at rs8018720) (Table 1; Manhattan plots and QQ-plots for overall samples are presented in Fig. 1, and regional association plots are presented in Supplementary Fig. 2). The associations at both novel loci were confirmed in the two independent in-silico replication cohorts (EPIC: $P = 1.21 \times 10^{-8}$ at rs10745742, $P = 5.24 \times 10^{-4}$ at rs8018720; SOCCS: $P = 0.03$ at rs10745742, $P = 0.04$ at rs8018720) with consistent direction of effect but slightly larger effect sizes and wider confidence intervals observed in SOCCS which could be due to the reduced sample size. When analyzing the two replication data sets together with the discovery data set, the $P$-values in pooled samples became more significant ($P_{pooled} = 2.10 \times 10^{-20}$ at rs10745742, $P_{pooled} = 1.11 \times 10^{-11}$ at rs8018720) (Table 1). We also found more than one distinct signal arising from variants at the *GC*, *CYP2R1*, and *AMDHD1* loci through conditional and joint analysis, but not for the *NADSYN1/DHCR7*, *CYP24A1*, and *SEC23A* loci where only one primary associated SNP was identified (Supplementary Table 2).

**SNP by dietary vitamin D intake interaction.** In addition to performing the marginal effect meta-analysis using all samples, we also tested a model with SNPs and dietary vitamin D intake as main effects and a term for their interaction in a subset of samples. Diet questionnaires, including vitamin D intake, were available for a subset of 13 cohorts and an additional 2 cohorts that were not included in the overall meta-GWAS analysis (a total of 15 cohorts, $N = 41,981$). We performed a GWAS explicitly allowing for an interaction between vitamin D intake and SNP genotypes, in which dietary vitamin D was coded as a continuous variable. We performed two tests: (i) a 1 degree-of-freedom interaction test between each SNP and vitamin D intake, and (ii) a 2 degree-of-freedom joint test of main genetic and interaction effects. However, for comparison purposes, we also performed, (iii) a standard test of marginal genetic effect after adjusting for vitamin D intake in the same sub-samples (Supplementary Fig. 3, and Supplementary Fig. 4). The marginal genetic effect analyses confirmed existing association signals at *GC* (lead SNP rs2282679, in complete linkage disequilibrium with the lead SNP rs3755967 identified from meta-GWAS using all individuals), *CYP2R1*, *NADSYN1/DHCR7* (lead SNP rs4944062, in complete linkage disequilibrium with the lead SNP rs12785878 identified from meta-GWAS using all individuals) and *CYP24A1*, as well as the novel association at *AMDHD1* ($P = 5.7 \times 10^{-9}$, Supplementary Table 3). The joint analysis also identified the five genes above, but with less significant $P$-values. For instance, the association at *AMDHD1* achieved only suggestive genome-wide significance ($P = 1.2 \times 10^{-7}$, Table 2). The interaction test did not identify any variants at genome-wide significance level. Among the 5 SNPs significant in marginal effect tests, the lead SNP in *CYP2R1* showed nominal significance for interaction with dietary vitamin D intake (rs10741657, $P = 0.028$), but no interactions were observed for the other SNPs (rs2282679, $P = 0.45$; rs4944062, $P = 0.74$; rs10745742, $P = 0.64$; rs17216707, $P = 0.46$). Repeating the analysis using a tertile coding for vitamin D instead of a continuous coding did not qualitatively change the results.

**SNP-heritability of 25-hydroxyvitamin D.** We further evaluated the SNP-heritability, defined as the heritability explained by GWAS SNPs of 25-hydroxyvitamin D, using LD score regression

**Table 1 Single nucleotide polymorphisms identified in genome-wide analyses for circulating 25-hydroxyvitamin D concentrations**

| Gene | SNP | Chromosome: Position | Effect/reference allele | Allele frequency | Meta-GWAS estimates | | |
|---|---|---|---|---|---|---|---|
| | | | | | Effect (Beta) | Standard Error | *P*-value |
| First stage discovery meta-GWAS (*N* = 79,366) | | | | | | | |
| GC | rs3755967 | 4:72828262 | T/C | 0.28 | −0.089 | 0.0023 | 4.74E-343 |
| NADSYN1/ DHCR7 | rs12785878 | 11:70845097 | T/G | 0.75 | 0.036 | 0.0022 | 3.80E-62 |
| CYP2R1 | rs10741657 | 11:14871454 | A/G | 0.4 | 0.031 | 0.0022 | 2.05E-46 |
| CYP24A1 | rs17216707 | 20:52165769 | T/C | 0.79 | 0.026 | 0.0027 | 8.14E-23 |
| AMDHD1 | rs10745742 | 12:94882660 | T/C | 0.4 | 0.017 | 0.0022 | 1.88E-14 |
| SEC23A | rs8018720 | 14:38625936 | C/G | 0.82 | −0.017 | 0.0029 | 4.72E-09 |
| Replication data set 1: samples collected by EPIC (*N* = 40,562) | | | | | | | |
| AMDHD1 | rs10745742 | 12:94882660 | T/C | 0.41 | 0.041 | 0.0071 | 1.21E-08 |
| SEC23A | rs8018720 | 14:38625936 | C/G | 0.83 | −0.032 | 0.0093 | 5.24E-04 |
| Replication data set 2: additional control samples collected by SOCCS (*N*=2195) | | | | | | | |
| AMDHD1 | rs10745742 | 12:94882660 | T/C | 0.37 | 0.045 | 0.021 | 0.03 |
| SEC23A | rs8018720 | 14:38625936 | C/G | 0.81 | −0.051 | 0.026 | 0.04 |
| Pooled analysis (discovery meta-GWAS + replication 1 + replication 2) (*N* = 122,123) | | | | | | | |
| AMDHD1 | rs10745742 | 12:94882660 | T/C | 0.39 | 0.019 | 0.002 | 2.10E-20 |
| SEC23A | rs8018720 | 14:38625936 | C/G | 0.82 | −0.019 | 0.0027 | 1.11E-11 |

In the pooled analysis, $P_{heterogeneity}$ = 0.003 for AMDHD1 rs10745742; $P_{heterogeneity}$ = 0.14 for SEC23A rs8018720.

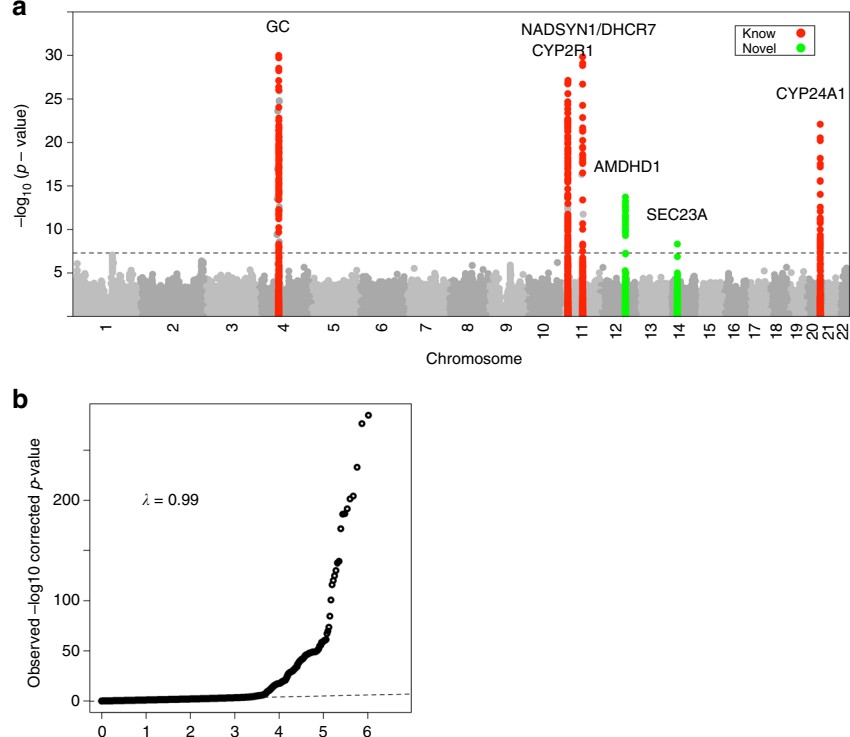

**Fig. 1** Genome-wide association of circulating 25-hydroxyvitamin D graphed by chromosome positions and −log10 *P*-value (Manhattan plot), and quantile-quantile plot of all SNPs from the meta-analysis (QQ-plot). **a** Manhattan plot: The *P*-values were obtained from the single stage fixed-effects inverse variance weighted meta-analysis. The *Y* axis shows −log$_{10}$ *P*-values, and the *X* axis shows chromosome positions. Horizontal gray dash line represents the thresholds of $P = 5 \times 10^{-8}$ for genome-wide significance. Known loci were colored coded as red, and novel loci were color coded as green. **b** QQ-plot: The *Y* axis shows observed −log$_{10}$ *P*-values, and the *X* axis shows the expected −log$_{10}$ *P*-values. Each SNP is plotted as a black dot, and the dash line indicates null hypothesis of no true association. Deviation from the expected *P*-value distribution is evident only in the tail area, with a lambda of 0.99, suggesting that population stratification was adequately controlled

(see Methods). The overall observed heritability of 25-hydroxyvitamin D estimated by using all common SNPs (2,579,296 after QC) was 7.54% (standard error (SE): 1.88%). After excluding genome-wide significant SNPs (533 SNPs with $P \leq 5 \times 10^{-8}$) from six loci and all SNPs within ±500 kb of those

loci, the heritability decreased to 4.70% (SE: 0.72%). The estimate further decreased to 1.73% (SE: 0.32%) after excluding all SNPs that reached nominal significance (156,675 SNPs with $P \leq 0.05$). These results indicate that common variants tagged by GWAS chips explain a modest fraction of overall variability in circulating

**Table 2 Results from the SNP-by-dietary vitamin D intake interaction analysis**

| Gene | SNP | Chromosome: Position | Effect/Reference Allele | Allele Frequency | SNP-by-dietary vitamin D intake Interaction analysis | | | | | |
| | | | | | Main Genetic Effect | | | Interaction Effect | P-value for interaction | P-value for joint test |
| | | | | | Effect (Beta_G) | Standard Error | Effect (Beta_Int) | Standard Error | | |
| First stage discovery meta-GWAS (N = 79,366) | | | | | | | | | | |
| GC | rs3755967 | 4:72828262 | T/C | 0.28 | −0.082 | 0.0042 | −2.01E−05 | 1.67E−05 | 0.23 | 2.92E−171 |
| | rs2282679* | 4:72827247 | T/G | 0.28 | 0.085 | 0.004 | 1.20E−05 | 1.60E−05 | 0.45 | 1.40E−187 |
| NADSYN1/DHCR7 | rs12785878 | 11:70845097 | T/G | 0.75 | 0.033 | 0.0039 | 6.61E−06 | 1.62E−05 | 0.68 | 3.52E−29 |
| | rs4944062* | 11:70864942 | T/G | 0.75 | 0.034 | 0.004 | 5.30E−06 | 1.60E−05 | 0.74 | 1.90E−31 |
| CYP2R1 | rs10741657 | 11:14871454 | A/G | 0.4 | 0.03 | 0.0035 | 3.21E−05 | 1.46E−05 | 0.028 | 2.23E−38 |
| CYP24A1 | rs17216707 | 20:52165769 | T/C | 0.79 | 0.025 | 0.0048 | 1.39E−05 | 1.88E−05 | 0.46 | 1.32E−14 |
| AMDHD1 | rs10745742 | 12:94882660 | T/C | 0.4 | 0.016 | 0.0036 | −7.05E−06 | 1.49E−05 | 0.64 | 1.20E−07 |
| SEC23A | rs8018720 | 14:38625936 | C/G | 0.82 | −0.013 | 0.0051 | −2.40E−05 | 2.06E−05 | 0.24 | 1.94E−05 |

* Top SNPs identified in the SNP-by-dietary vitamin D intake interaction analysis, performed in a subset of individuals. For GC and NADSYN1/DHCR7, the top SNPs identified through the marginal effect regression meta-analysis using all individuals were in high linkage disequilibrium with the top SNPs identified through the SNP-by-dietary vitamin D intake interaction analysis using a subset of individuals ($r^2$ for rs3755967 and rs2282679: 1.0; $r^2$ for rs12785878 and rs4944062: 1.0). Beta_G indicates the main effect of the SNP, Beta_Int indicates the interaction effect of SNP-by-dietary vitamin D intake

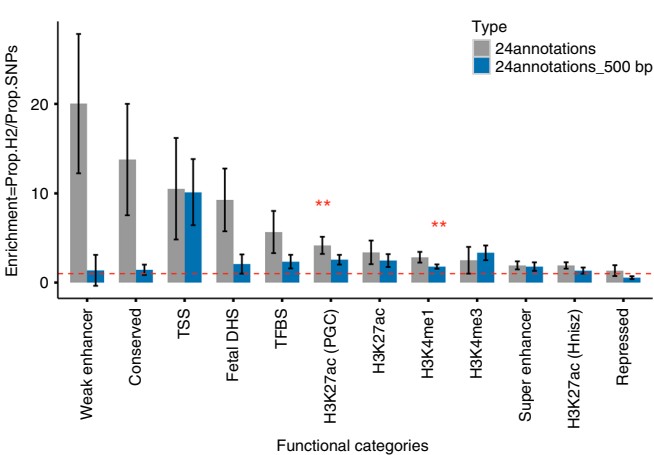

**Fig. 2** Heritability enrichment of the top 12 genomic functional elements. We partitioned the SNP-heritability of serum 25-hydroxyvitamin D concentrations into 24 publicly available genomic functional elements using LD-score regression. We plotted the enrichment (Y axis) for each of the 12 top annotations (as shown in X axis) into a bar chart. Gray bars and blue bars represent the annotations with and without the 500 base-pair windows. The height of each bar represents magnitude of enrichment. Significant estimates of enrichment that passed Bonferroni corrections (P-value for enrichment <0.05/24) are marked with double stars. TSS transcription start sites, DHS DNase I hypersensitive sites, TFBS transcription factor binding sites, Repressed repressed regions

25-hydroxyvitamin D, and that an appreciable proportion of this SNP-heritability is explained by the six genetic regions of associated SNPs identified through GWAS.

**Partitioning the total heritability of 25-hydroxyvitamin D.** We next partitioned the heritability by functional elements using baseline model with 24 publicly available annotations (see Methods), and observed large and significant enrichment for several functional categories (Fig. 2, Supplementary Table 4). For example, we found the largest enrichment in weak enhancers, with 2.1% of SNPs explaining 42.3% of the overall heritability (20-fold enrichment, $P = 0.02$), followed by conserved regions (13.8-fold enrichment, $P = 0.03$), open chromatin (as reflected by

DHS, 8.5-fold enrichment, $P = 0.02$), transcription factor binding sites (5.7-fold enrichment, $P = 0.048$), super-enhancers (1.9-fold enrichment, $P = 0.04$), and all four histone marks were enriched (both versions of H3K27ac (one version processed by Hnisz et al., and another version used by the Psychiatric Genomics Consortium (PGC), H3K4me1, H3K4me3 (500 bp), H3K29ac (500 bp)). We also observed depletion for repressed regions (0.06-fold enrichment, $P = 0.006$). However, none of those annotations withstood multiple-testing corrections (Bonferroni corrected P-threshold: 0.05/24) except for the active enhancer histone mark H3K27ac (PGC) (4.2-fold enrichment, $P = 8 \times 10^{-4}$) and H3K4me1 (1.8-fold enrichment, $P = 0.0019$).

We subsequently performed cell-type-specific analysis by using 10 broad cell-type groups. As shown in Table 3, the top three enrichments were in the immune and hematopoietic tissues (4.3-fold enrichment, $P = 2.2 \times 10^{-5}$), gastrointestinal tissues (4.4-fold enrichment, $P = 0.0017$), and CNS (3.6-fold enrichment, $P = 0.0039$). There was also significant enrichment for liver, kidney, and connective and bone tissues, but these results did not survive multiple-testing corrections. When further analyzing 220 cell-type-specific annotations, we observed the most significant enrichment in CD19 cells (approximately 8-fold enrichment, $P \sim 0.001$), followed by CD20 cells (6.4-fold enrichment, $P = 0.003$) and CD3 cells (7.8-fold enrichment, $P = 0.01$) (Supplementary Data 1).

**Genetic correlations between 25-hydroxyvitamin D and traits.** We continued to assess the genetic correlation between 25-hydroxyvitamin D and each of the 37 traits with publicly available GWAS summary statistics data (Supplementary Table 5). None of the genetic correlations remained significant after Bonferroni correction (corrected P-threshold: 0.05/37, Fig. 3). Without multiple-testing correction, there were some correlations with nominal statistical significance. For example, ever smoking ($r_g$ (SE): −0.17 (0.073), $P = 0.019$), primary biliary cirrhosis ($r_g$ (SE): −0.18 (0.076), $P = 0.019$) and BMI adjusted waist-hip-ratio ($r_g$ (SE): −0.10 (0.050), $P = 0.042$) were observed to be inversely correlated with 25-hydroxyvitamin D; whereas lung function ($r_g$ (SE): 0.14 (0.046), $P = 0.0036$) showed a positive correlation with 25-hydroxyvitamin D. Subsequent directional genetic correlation analysis did not reveal any apparent putative causal relationship of 25-hydrovyvitamin D with other traits, except for a potential link between 25-hydroxyvitamin D and

**Table 3 Heritability enrichment of ten grouped cell types**

| Category | Proportion of SNPs (%) | Proportion of $h_g^2$ (%) | Enrichment (standard errors) | P-value |
|---|---|---|---|---|
| Kidney | 4.26 | 27.27 | 6.4 (2.44) | 0.027 |
| Liver | 7.22 | 37.68 | 5.22 (1.55) | 0.01 |
| Gastrointestinal | 16.77 | 72.88 | 4.35 (0.97) | **0.0017** |
| Immune and hematopoietic | 23.34 | 100.17 | 4.29 (0.76) | **2.20E-05** |
| Central nervous system | 14.88 | 54.09 | 3.64 (0.87) | **0.0039** |
| Cardiovascular | 11.11 | 35.74 | 3.22 (1.26) | 0.078 |
| Connective tissue/bone | 11.5 | 35.65 | 3.1 (1.04) | 0.037 |
| Adrenal/pancreas | 9.36 | 26.17 | 2.8 (1.31) | 0.18 |
| Other | 20.27 | 56.68 | 2.8 (0.98) | 0.076 |
| Skeletal Muscle | 10.38 | 14.29 | 1.38 (1.25) | 0.76 |

Black bold font indicates significant P-values after multiple corrections ($P < 0.05/10$)

HDL (Supplementary Table 6). However, with only six 25-hydroxyvitamin D associated SNPs included in the analysis, we consider an overall null directional correlation as our main finding, and further well-designed large-scale Mendelian randomization analyses are warranted.

Finally, we analyzed the 220 cell-type-specific annotations in each of the 37 traits and compared the cell-type-specific enrichments for 25-hydroxyvitamin D to the enrichments for these traits. The enrichment pattern for 25-hydroxyvitamin D differed notably from the patterns for psychiatric diseases and metabolic related traits. Psychiatric diseases showed enrichment for histone marks specific to CNS cell types, and metabolic diseases showed enrichment for gastrointestinal cell types, while these annotations were depressed in 25-hydroxyvitamin D. Conversely, 25-hydroxyvitamin D showed similar patterns with autoimmune inflammatory diseases, where multiple immune cell types were enriched. We consistently observed that 25-hydroxyvitamin D was clustered with autoimmune diseases (Supplementary Fig. 5).

## Discussion

Vitamin D inadequacy has been linked to many diseases such as cancer, autoimmune disorder and cardiovascular conditions in addition to musculoskeletal diseases, which has led to substantial interest in the determinants of vitamin D status, especially its genetic components. We have performed a large 25-hydroxyvitamin D meta-GWAS involving 31 studies with a total of 79,366 individuals. Our results recapitulated several previously reported findings. First of all, we confirmed the role for common genetic variants in regulation of circulating 25-hydroxyvitamin D concentrations. Our study validated three loci, GC, NADSYN1/DHCR7, and CYP2R1, all were established 25-hydroxyvitamin D risk loci identified through two earlier GWASs[8,11]. In addition, we were able to confirm the association of a locus containing CYP24A1 with 25-hydroxyvitamin D concentrations using our large sample size, which highlights the importance of this protein in the degradation of vitamin D molecule, by catalyzing hydroxylation reactions at the side chain of 1,25-dihydroxyvitamin D, the physiologically active form (hormonal form) of vitamin D. Significant finding at this locus was only shown in the pooled analyses involving both discovery and replication samples in an earlier GWAS[8].

We extended previously reported findings by identifying two additional new loci. SEC23A (Sec23 Homolog A, coat protein complex II (COPII) component) encodes a member of SEC23 subfamily. In eukaryotic cells, secreted proteins are synthesized in the endoplasmic reticulum (ER), packaged into COPII-coated vesicles, and traffic to the Golgi apparatus. As part of COPII complex, SEC23 plays a role in promoting ER-Golgi protein trafficking. SEC23A mutations have been reported to cause craniolenticulosutural dysplasia, a disease characterized by craniofacial and skeletal malformation such as delay in closure of fontanels, sutural cataracts and facial dysmorphisms, due to defective collagen secretion[13,14]. The second novel locus is AMDHD1 (amidohydrolase domain containing 1). This gene encodes an enzyme involved in the histidine, lysine, phenylalanine, tyrosine, proline and tryptophan catabolic pathway. Mutations in AMDHD1 are found to be associated with atypical lipomatous tumor, a cancer of connective tissues that resemble fat cells[15].

Our SNP-heritability results suggest that 25-hydroxyvitamin D has a modest overall heritability due to common genome-wide SNPs of 7.5%, and that an appreciable proportion (2.84% out of 7.5%, i.e., 38%) of this total could be explained by known genetic regions identified through GWAS. Our findings are in line with a previous published report (by Hiraki et al.[12]) which estimated the variance in circulating 25-hydroxyvitamin D explained by SNPs in a total of 5575 individuals[12]. According to that report, by employing a linear mixed model fitting the additive genetic matrix created from all genotyped and imputed SNPs, the proportion of variance explained was 8.9%; by employing a polygenic score approach comprised of the then GWAS-discovered SNPs (GC, CYP2R1, DHCR7/NADSYN1), the proportion of variance explained was 5%. Both of these estimates were close to ours. In Hiraki et al., the known 25-hydroxyvitamin D associated environmental factors such as age, BMI, season of blood drawn, vitamin D dietary intake, vitamin D supplement intake, region of residence and ethnicity, explained ~18% of the observed variance[12]. Our results, in agreement with these findings, suggest that although there appears to be some polygenic signals outside of the identified regions, the remaining common effects may be small. There also may be low frequency variants with larger effects that were not investigated here. For example, while this paper was under review, a related study identified low-frequency (MAF = 2.5%) synonymous coding variant rs117913124_A at CYP2R1 conferring a large effect on 25-hydroxyvimtain D levels, which was four times greater in magnitude and independent of a previously described association for a common variant (rs10741657) near CYP2R1[16].

Results of twin and familial studies have revealed a substantial genetic basis in the variability of circulating 25-hydroxyvitamin D levels, with estimates of heritability reaching as high as 86%[9,10,17–19]. These estimates, however, seem to be influenced by environmental conditions. For example, in a study conducted by Orton et al. with 40 monozygotic and 59 dizygotic twin pairs, bloods were collected at the end of winter and a heritability of 77% was reported[10]. Similarly, the study conducted by Karohl et al. with 310 monozygotic and 200 dizygotic male twins observed a heritability of 70% during winter, whereas in summer, serum 25-hydroxyvitamin D concentrations appeared to be

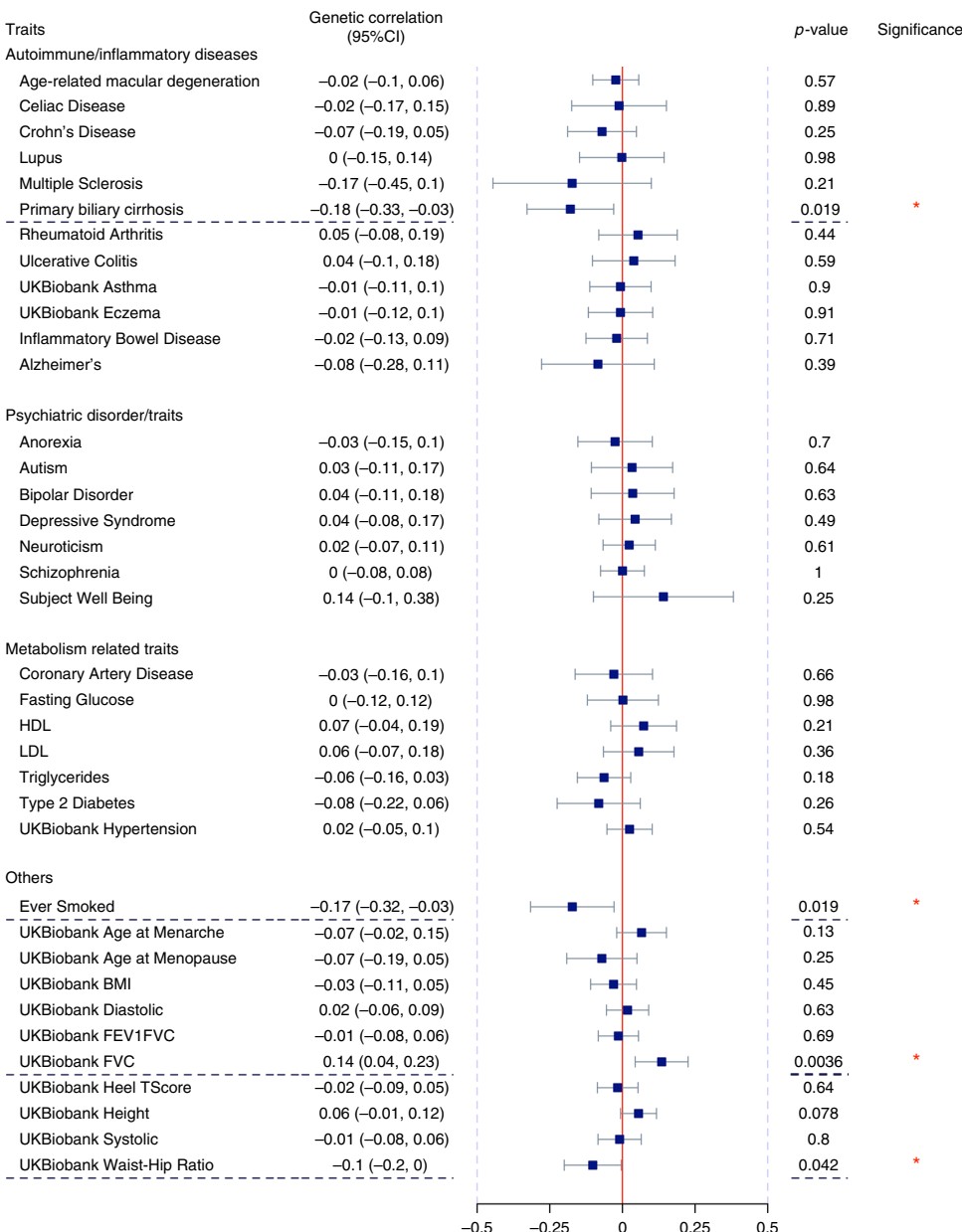

**Fig. 3** Genetic correlations between 25-hydroxyvitamin D and 37 traits. We collected GWAS summary statistics of 37 diseases and traits spanning a wide range of phenotypes (autoimmune inflammatory diseases, psychiatric disorders, metabolic traits, and anthropometric index) from publicly available resources, and estimated their shared genetic similarities with serum 25-hydroxyvitamin D levels. We plotted the genetic correlation together with 95% confidence intervals using a blue square and gray horizontal lines. Red vertical line indicates no genetic correlation ($r_g = 0$). Statistical significance was defined as $P$-value <0.05. None of the pairwise correlations passed Bonferroni corrections

entirely determined by non-genetic factors (heritability: 0%)[9]. Comparable estimates were also identified in a slightly larger study conducted by Mills et al. (winter: 90% *vs.* summer: 56%)[18]. Consistent with season dependency, sex differences were also observed (males: 86% *vs.* females: 17%)[17]. While these estimates should be treated with caution due to small samples and related imprecision, they confirm the substantial variation in 25-hydroxyvitamin D levels by season (as shown previously[20]) and illustrate that heritability estimates derived from a homogenous source may be highly inflated. In a relevantly well-powered twin study with a total of ~2100 female twins, the heritability of 25-hydroxyvitamin D was calculated to be 40%, indicating a larger proportion of variance explained by non-genetic factors[21]. Heritability estimates obtained using GWAS SNPs have typically been found to be approximately half of those from classical twin

studies[9,10], but our estimate of 7.5%, calculated using common genome-wide SNPs, is far lower than reported heritability from twin and family based studies. In addition to potentially inflated estimates from twin studies, the difference may reflect the proportion of heritability explained by rare SNPs or structural variants that were not included in our data, and the potential gene-gene interactions that remain to be identified. The combination of our samples from all seasons is also likely to decrease the probability of finding genetic variants, and hence deflate heritability estimates.

Through partitioning the SNP-heritability of serum 25-hydroxyvitamin D levels, we observed a significant enrichment in immune and hematopoietic tissues; likewise, the cell-type-specific analysis revealed clustering of 25-hydroxyvitamin D and autoimmune diseases, indicating that these traits share a majority

of common cell types. The link between vitamin D deficiency and increased risk for autoimmune inflammatory diseases has long been recognized by epidemiological investigations[22,23]. Although the underlying mechanisms remain unclear, it is now evident that vitamin D is involved in many biological processes that regulate both innate and adaptive immune responses, through ligand-receptor binding, activation, interaction with response elements in the promoter regions of different genes, and eventually lead to functional changes in a wide variety of immune cells including Th1, Th2, Th17, T regulatory and natural killer T cells[22,23]. The shared cell type enrichments between vitamin D and autoimmune diseases observed in our study, further suggest that vitamin D not only affects autoimmune diseases through its direct effect (as a ligand), but also through their shared genetic etiology. Thus, individuals with vitamin D deficiency may be more susceptible to these disorders, both because of environmental and genetic influences.

Our genome-wide interaction analysis between genetic variants and dietary intake of vitamin D did not identify new signals. All significant associations observed in the joint test of main genetic and interaction effects were of equal or higher significance level (i.e., lower P-values) in the GWAS of marginal genetic effect performed in the same individuals, indicating no major contribution of interaction effects at these loci. Indeed, only one of the top 5 loci from the overall marginal GWAS showed nominally significant interaction effect, and none passed Bonferroni corrections. While smaller gene-diet interaction effects remain to be discovered, our results provide some evidence against large interactions between common SNPs and dietary vitamin D intake. Still, one cannot completely rule out the possibility of interaction, but only conclude that genetic effects appear stable within vitamin D intake range in the populations studied. Indeed, as for any gene-environment interaction tests, statistical power is highly dependent on the variance of exposure in the samples analyzed[24], and interactions would remain unobserved if the exposure is homogeneous among individuals. Also, we were not able to capture vitamin D supplementation adequately to include this in the dietary intake variable, and were not able to estimate sunlight exposure as a source of vitamin D production in the skin.

Serum 25-hydroxyvitamin D concentrations are mainly determined by modifiable environmental factors, and contrary to estimates from previous twin studies, our large-scale analyses suggest a SNP-heritability rate that is relatively modest in magnitude when considering common variants. Our study also showed that common genetic variants are unlikely to have a strong modifying effect on increases in 25-hydroxyvitamin D following typical dietary intakes, suggesting that consideration of genetic background is not required when determining population based vitamin D intake recommendations. However, our results support the role of vitamin D in immunological diseases as we observed from cell-type-specific analysis for clustering of vitamin D and autoimmune diseases, and the evidence for signal enrichment for immune and hematopoietic tissues. These findings are in line with previous Mendelian Randomization studies which found a putative causal association between vitamin D and autoimmune diseases such as multiple sclerosis[1,2] and type 1 diabetes[25]. The additional genetic instruments identified by our results could also be used in future Mendelian Randomization analyses of the association between vitamin D and complex traits.

## Methods

**Study cohorts.** We expanded our previous SUNLIGHT consortium GWAS, and undertook a large, multicenter, genome-wide association study of 31 cohorts in Europe, Canada and USA. Our first stage discovery meta-analysis consisted of 79,366 samples of European descent drawn from 31 epidemiological cohorts. Among those 31 cohorts, ten were used as discovery and in-silico replication

samples in our previous GWAS publication (the 1958 British Birth Cohort (1958BC), the Cardiovascular Health Study (CHS), the Framingham Heart Study (FHS), the Gothenburg Osteoporosis and Obesity Determinants study (GOOD), the Health, Aging, and Body Composition study (Health ABC), the Indiana Women cohort, the North Finland Birth Cohort 1966 (NFBC1966), the Old Order Amish Study (OOA), the Rotterdam Study (RS), and the TwinsUK), and an additional 21 cohorts were included for the current analysis (the Alpha-Tocopherol, Beta-Carotene Cancer Prevention Study (ATBC), the Atherosclerosis Risk in Communities Study (ARIC), the AtheroGene registry, B-vitamins for the Prevention Of Osteoporotic Fractures (B-PROOF), the Epidemiology of Diabetes Interventions and Complications (EDIC), the Case-Control Study for Metabolic Syndrome (GenMets), the Helsinki Birth Cohort Study (HBCS), the Health Professional Follow-up Study (HPFS, nested a coronary heart disease case-control study), the Invecchiare in Chianti Study (InChianti), the Cooperative Health Research in the region Augsburg (KORA), the Leiden Longevity Study (LLS), the Ludwigshafen Risk and Cardiovascular Health Study (LURIC), the Multi-Ethnic Study of Atherosclerosis (MESA), the Nijmegen Biomedische Studie (NBS), the Nurses' Health Study (NHS, nested a breast cancer case-control study, and a type2 diabetes case-control study), the Orkney Complex Disease Study (ORCADES), the Prostate, Lung, Colorectal, and Ovarian Cancer Screening Trial (PLCO), the PROspective Study of Pravastatin in the Elderly at Risk (PROSPER), the Study of Health in Pomerania (SHIP), the Scottish Colorectal Cancer Study (SOCCS), the Cardiovascular Risk in Young Finns Study (YFS), and more samples from the RS (RSI, RSII, and RSIII)). Full descriptions of all participating cohorts, details of genotyping platforms used, number of SNPs, and the measurements of serum 25-hydroxyvitamin D concentrations in each cohort are shown in Supplementary Table 1 and Supplementary Note 1. Written informed consent was obtained from all participants in the included cohorts, and the study protocols were reviewed and approved by local institutional review boards.

**Power calculation.** Our large sample size provided good statistical power for association analysis. At the genome-wide significance threshold of $5 \times 10^{-8}$, with a discovery sample size of 75,000, our study had 85% power to detect a genetic variant (single nucleotide polymorphism, SNP) accounting for 0.06% of the total variance in serum 25-hydroxyvitamin D concentrations, and 99% power to detect a variant that explained 0.1% of the total variance. We also had power to detect gene-environment interaction effects even smaller than the observed marginal effects. In the case where a SNP has no marginal effect on circulating 25-Hydroxyvitamin D concentrations (and so could not have been discovered via the marginal GWAS), we had 80% power to detect an interaction that explained 0.07% of the total variance in 25-hydroxyvitamin D concentrations.

**Association analysis.** Genome-wide analyses were performed within each cohort according to a uniform analysis plan. We fit additive genetic models using linear regression on natural-log transformed 25-hydroxyvitamin D, and adjusted the models for month of sample collection (12 categories), age, sex, and body mass index, and principal components capturing genetic ancestry. Further adjustments included cohort-specific variables, such as geographical location and assay batch, where relevant. For participating studies with a case-control design, we analyzed cases and controls separately. We performed a fixed-effects inverse variance weighted meta-analysis across the contributing cohorts, as implemented in the software METAL[26], with control for population structure within each cohort and quality control thresholds of minor allele frequency (MAF) > 0.05, imputation info score > 0.8, Hardy-Weinberg equilibrium (HWE) > $1 \times 10^{-6}$, and a minimum of two studies and 10,000 individuals contributing to each reported SNP-phenotype association. We regarded P-values < $5 \times 10^{-8}$ as genome-wide significant.

**Replication study.** We replicated the identified novel loci in two independent data sets for which genotype data were available: the European Prospective Investigation into Cancer and Nutrition (EPIC) study with 40,562 individuals across two nested case-control studies (EPIC-InterAct and EPIC-CVD) and the cohort-wide EPIC-Norfolk study (Supplementary Note 1); and a cohort of 2195 individuals (all controls) additionally collected as part of the SOCCS that were not included in our discovery stage. As for the phenotype, EPIC individuals were assayed for plasma 25-hydroxyvitamin $D_3$ and SOCCS individuals were assayed for total 25-hydroxyvitamin D. We performed the association analysis in a similar manner, adjusted for age, sex, time of sample collection, and study center where relevant. We regarded P-value < 0.05 in the replication samples, and P-value < $5 \times 10^{-8}$ in the pooled analysis as successful replication.

**Conditional analysis.** After identifying the primary associated variant at each locus selected according to the strength of its association, we further tested whether there were any other SNPs significantly associated with 25-hydroxyvitamin D after accounting for the effect of lead SNP. We thus performed a stepwise model selection procedure for those chromosomes where a significant variant was previously identified. We started with the most significantly associated SNP, scanning through the whole chromosome, selecting additional independently associated SNPs using a stepwise procedure, one at a time, based on their conditional P-values. Finally, we fit all selected SNP into one model to estimate their joint effects.

We used GCTA-COJO software to accommodate our summary level GWAS data[27], and the Cancer Genetic Marker of Susceptibility (CGEMS) GWAS with 2287 individuals of European descent and 2,543,887 genotyped and imputed (HapMap22) SNPs as reference panel.

**SNP-by-diet interaction**. We performed a genome-wide association screening of circulating 25-hydroxyvitamin D while accounting for potential interaction effect between SNP and dietary vitamin D intake. Our tests incorporating gene-diet interaction were based on the following model:

$$\ln(25(OH)D) = \beta_0 + \beta_1 \times G + \beta_2 \times E + \beta_3 \times G \times E + \beta_Z \times Z$$

where $G$ is a SNP that was coded additively, $E$ is the raw vitamin D intake, measured on a continuous scale. The parameters $\beta_0$, $\beta_1$, $\beta_2$, and $\beta_3$ are the intercept, the main effect of SNP, the main effect of dietary vitamin D intake, and the interaction effect between $G$ and $E$. The model also included the same covariates $Z$ as for the marginal effect screening, the effects of which were captured in the parameter $\beta_Z$. We considered both a standard 1 degree-of-freedom test of interaction effect (i.e., null hypothesis of $\beta_3 = 0$), and a joint 2 degree-of-freedom test of main genetic effect plus gene-by-diet interaction (i.e., null hypothesis of $\beta_1 = 0$ and $\beta_3 = 0$). For comparison purposes, we also considered a model adjusting for vitamin D intake but not modeling interaction (i.e., not including the $\beta_3 \times G \times E$ term) using the same subset of individuals.

Vitamin D intake was available for 15 cohorts on a total of 41,981 individuals. It included both the population based cohorts (ARIC, 1958BC, B-PROOF, FHS, Health ABC, MESA, NFBC, RS, RS III, and YFS as part of the overall Meta-GWAS, plus two additional cohorts, the Prospective Investigation of the Vasculature in Uppsala Seniors (PIVUS), and the Uppsala Longitudinal Study of Adult Men (ULSAM), genotyped on custom array that were not included in the overall meta-GWAS but were included in this SNP-by-diet interaction analysis), and case-control studies (HPFS (HPFS_CHD), NHS (NHS_BRCA, NHS_T2D), and SOCCS). For the latter studies, all analyses were performed separately in cases and controls. The aforementioned interaction model was applied to each of the included cohorts, and study-specific results were meta-analyzed using inverse-variance weighted sum of effect estimates as implemented in METAL[26]. For the 2 degree-of-freedom test we used joint framework described in two previous published papers[28,29]. Quality control filtering was performed on each study before meta-analysis. Only SNPs with imputation info score > 0.8, MAF > 0.05, HWE > 1×10⁻⁶, and a minimum of total sample size in the meta-analysis > 10,000 were retained.

**Linkage Disequilibrium score regression**. We performed linkage disequilibrium score regression (LDSC) analysis to estimate the SNP-heritability of serum 25-hydroxyvitamin D concentrations[30,31]. This method is based on a validated relationship between LD score and $\chi^2$-statistics:

$$E\left[\chi_j^2\right] \approx \frac{N_j h_g^2}{M} l_j + 1$$

where $E\left[\chi_j^2\right]$ denotes the expected $\chi^2$-statistics for the association between outcome and SNP $j$, $N_j$ is the study sample size available for SNP $j$, $M$ is the total numbers of variants and $l_j$ denotes the LD score of SNP j defined as $l_j = \sum r^2(j,k)$. LDSC calculates heritability using only summary-level data instead of individual genotypes, and is computationally cost effective at large sample sizes. We used the summary statistics from 25-hydroxyvitamin D meta-GWAS results, with SNPs available in at least 2 studies and a sample size of at least 10,000. We first analyzed the SNP-heritability by using (1) SNPs across the entire genome that passed quality control; (2) SNPs excluding the top associations (SNPs reaching genome-wide significance, $P \leq 5 \times 10^{-8}$), as well as all SNPs within ±500 kb of the top hits in the region; and (3) SNPs excluding the nominally significant associations ($P \leq 0.05$). We subsequently partitioned the heritability through three different models: (1) a full baseline model including the 24 publicly available main annotations that are not specific to any cell type, the 500-bp windows around each annotation, as well as 100-bp windows around chromatin immunoprecipitation and sequencing peaks (ChIP-seq) when appropriate. This resulted in a total of 52 overlapping functional categories in the full baseline model; (2) a cell-type-specific model including 10 cell type groups: adrenal and pancreas, central nervous system (CNS), cardiovascular, connective and bone, gastrointestinal, immune and hematopoietic, kidney, liver, skeletal muscle, and other; and (3) a cell-type-specific model including 220 cell-type-specific annotations for the four histone marks with putative enhancer or promoter functions, H3K4me1, H3K4me3, H3K9ac, and H3K27ac.

Details of the 24 publicly available annotations, the 220 cell-type-specific annotations, as well as the 10 cell type groups were described by Finucane et al.[31]. Briefly, the 24 annotations included coding, UTR (3′UTR and 5′UTR), promoter and intronic regions, acquired from the UCSC Genome Browser[32] and post-processed by Gusev et al.[33]; the three histone marks (mono-methylation (H3K4me1) of histone H3 at lysine 4, tri-methylation (H3K4me3) of histone H3 at lysine 4, and acetylation of histone H3 at lysine 9 (H3K9ac) processed by Trynka et al.[34–36] and two versions of acetylation of histone H3 at lysine 27 (H3K27ac, one version processed by Hnisz et al.[37], another version used by the Psychiatric

Genomics Consortium (PGC)[38]; open chromatin, as reflected by DNase I hypersensitivity sites (DHSs and fetal DHSs)[33], obtained as a combination of Encyclopedia of DNA Elements (ENCODE) and Roadmap Epigenomics data, and processed by Trynka et al.[36]; combined chromHMM and Segway predictions obtained from Hoffman et al.[39], which leverage on many annotations to partition the genome into seven underlying chromatin states (the CCCTC-binding factor (CTCF), promoter-flanking, transcribed region, transcription start site (TSS), strong enhancer, weak enhancer, and the repressed region); regions that are conserved in mammals, provided by Lindblad-Toh et al.[40] and post-processed by Ward and Kellis[41]; super-enhancers, which are large groups of putative enhancers with high levels of activity, provided by Hnisz et al.[37]; FANTOM5 enhancers mapped by using cap analysis of gene expression in the FANTOM5 panel of samples, obtained from Andersson et al.[42]; digital genomic footprint (DGF) and transcription factor binding site (TFBS) annotations downloaded from ENCODE[35] and post-processed by Gusev et al.[33]. We included 500-bp windows around each of the 24 main annotations in the baseline model, and 100-bp windows around ChIP-seq when appropriate, to prevent upward bias of estimates generated by enrichment in the nearby regions.

In addition to the baseline model using 24 main annotations, we also performed cell-type-specific analyses using annotations of the four histone marks (H3K4me1, H3K4me3, H3K9ac and H3K27ac). Each cell-type-specific annotation corresponds to a histone mark in a single cell type (for example, H3K27ac in adipose nuclei tissues), and there was a total of 220 such annotations. We further subdivided these 220 cell-type-specific annotations into 10 categories by aggregating the cell-type-specific annotations within each group (for example, SNPs related with any of the four histone modifications in any hematopoietic and immune cells were considered as one big category). When generating the cell-type-specific models, we added each annotation individually (one at a time) to the baseline model, creating separate models to control for overlap with the genomic functional elements in the full baseline model but not overlap with the other cell types.

We additionally assembled the summary statistics from GWAS of 37 traits or diseases performed in individuals of European descent, which are publicly available[38,43–55] or applied from the UK Biobank. These studies span a wide range of phenotypes, from anthropometric indices such as height, weight, BMI, to mental disorders (for example depressive syndrome and schizophrenia) to autoimmune and inflammatory diseases (for example rheumatoid arthritis and celiac diseases). We calculated the pairwise genetic correlation ($r_g$, cross trait heritability) between 25-hydroxyvitamin D and each of the 37 traits. We further conducted the same cell-type-specific analysis for each trait, and plotted beta-coefficient z-score matrix, constructed from the total 220 annotations by 37 traits, into four heat-maps based on the four histone marks.

Finally, in addition to the genetic correlation analysis which reflects shared genetic factors across different traits but does not inform direction, we also attempted to identify directions of such correlation using an algorithm proposed by Pickrell et al.[56]. The method adopts a similar intuition as the Mendelian Randomization approach, where, if a trait X influences trait Y, then SNPs influencing X should also influence Y, and the SNP-specific effect sizes for the two traits should be correlated. Further, since Y does not influence X, but could be influenced by mechanisms independent of X, genetic variants that influence Y do not necessarily influence X. Based on this intuition, the method proposes two "causal" models and two "non-causal" models, and calculates the relative likelihood ratio of the best non-causal model compared to the best causal model. We determined significant SNPs for each given trait by selected genome-wide significant ($P < 5 \times 10^{-8}$) SNPs and pruned the numbers based on their LD-pattern in the European populations in Phase1 of 1000 Genome Project. We scanned through all pairs of 25-hydroxyvitamin D and traits to identify directional correlations. We consider pairs of traits with likelihood ratio $_{\text{non-causal vs. causal}} <$ 0.05 as having evidence of directional correlations.

**Data availability**. The GWAS summary statistics on serum circulating vitamin D concentrations is available at dbGap https://drive.google.com/drive/folders/0BzYDtCo_doHJRFRKR0ltZHZWZjQ; all relevant data are available from the authors upon request.

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

## Acknowledgements

A full list of acknowledgements can be found in Supplementary Note 2.

## Author contributions

C.P., E.H., M.I.M., E.A.S., P.L.L., E.B., D.A., S.J.W., N.D.F., W.H., L.C.P.G.M.D.G., N.M.V.S., N.v.V., J.B.R., B.K., T.J.W., D.P.K., R.S.V., C.O., M.L., J.G.E., S.B.K., M.B., E.S., I.H.D.B., J.I.R., S.S.R., M.J., P.K., J.F.W., L.L., K.M., L.Z., H.C., E.T., S.M.F., M.G.D., T.L., M.K., O.T.R., V.M., M.A.I., J.W.J., N.J.W., C.L., N.G.F., K.K., A.B., and J.D. designed and managed individual studies. C.P., E.H., M.I.M., E.A.S., P.L.L., D.A., S.J.W., W.H., N.M.V.S., A.E., J.B.R., R.S.V., C.O., L.V., M.L., J.G.E., S.B.K., M.B., D.V.H., I.H.D.B., J.I.R., S.S.R., J.F.W., L.L., E.I., K.M., H.C., E.T., S.M.F., M.G.D., T.L., M.K., O.T.R., V.M., M.A.I., N.S., J.W.J., N.J.W., C.L., N.G.F., K.K., A.B., and J.D. collected data. E.B., A.E., C.O., M.L., Y.L., M.B., E.M.K., J.I.R., S.S.R., J.F.W., L.L., E.I., K.M., S.T., S.M.F., M.G.D., T.L., L.L., J.W.J., N.J.W., C.L., A.B., and J.D. performed the genotyping. C.P., D.B., E.H., M.I.M., W.T., E.B., N.M.V.S., A.E., J.D., C.O., L.V., M.L., K.K.L., J.D., E.M.K., J.I.R., S.S.R., J.F.W., C.H., E.I., K.M., S.T., A.G.U., F.R., L.Z., S.M.F., M.G.D., A.M.V., T.L., L.L., M.A.I., N.S., N.J.W., C.L., A.B., and J.D. prepared the genotype data. D.B., E.H., M.I.M., E.A.S., P.L.L., A.E., J.B.R., S.B., R.S.V., C.O., L.V., M.L., J.G.E., D.K.H., D.V.H., E.M.K., I.H.D.B., A.C.W., J.F.W., L.L., K.M., M.C.Z., A.G.U., F.R., L.Z., E.T., S.M.F., M.G.D., A.M.V., E.T., L.L., N.J.W., C.L., N.G.F., A.B., and J.D. prepared the phenotype data. D.B., E.H., M.I.M., J.B.R., T.J.W., D.P.K., Y.H., C.L., A.C.W., C.R.C., P.F.O., M.J., X.J., H.A., N.J.W., C.L., N.G.F., A.B., and J.D. developed the analysis plan. A.Z., D.B., E.H., M.I.M.,

P.L.L., W.T., L.C.P.G.M.D.G., N.v.V., A.E., J.B.R., B.K., T.J.W., J.D., D.P.K., Y.H., C.L., D.K.H., I.H.D.B., A.C.W., J.I.R., S.S.R., C.R.C., P.F.O., M.J., X.J., H.A., M.C.Z., A.G.U., F.R., L.B., N.J.W., C.L., and N.G.F reviewed the analysis plan. D.B., E.H., M.I.M., L.Y., W.T., A.E., J.B.R., Y.H., C.L., Y.Z., K.K.L., J.D., A.C.W., P.F.O., A.C., X.J., H.A., P.K.J., C.H., S.T., L.B., L.Z., E.T., E.T., L.L., J.Z., and M.T analyzed the data. D.B., Y.H., P.F.O., and H.A performed the meta-analysis. Y.H. and X.J. performed the pathway and other analyses. C.P., E.H., M.I.M., E.A.S., P.L.L., L.Y., W.T., E.D.M., E.B., L.C.P.G.M.D.G., N.v.V., A.E., J.B.R., T.J.W., J.D., D.P.K., Y.H., P.F.O., M.J., X.J., H.A., E.I., K.M., S.T., M.C.Z., A.G.U., F.R., L.B., L.Z., E.T., N.J.W., and N.G.F interpreted results. E.H., T.J.W., D.P.K., L.A.C., R.S.V., Y.H., I.H.D.B., J.I.R., S.S.R., M.J., P.K., A.G.U., F.R., N.S., and J.W.J. supervised the overall study design. E.H., J.B.R., T.J.W., D.P.K., Y.H., P.F.O., M.J., X.J., and H.A. wrote the manuscript. C.P., E.H., M.I.M., E.A.S., P.L.L., L.Y., W.T., E.D.M., E.B., D.A., S.J.W., N.D.F., W.H., L.C.P.G.M.D.G., N.M.V.S., N.v.V., J.B.R., B.K., T.J.W., J.D., D.P.K., D.K., S.B., R.S.V., Y.H., C.L., Y.Z., C.O., L.V., M.L., J.G.E., M.K.S., D.K.H., M.P., M.J.E., E.M.K., S.P., I.H.D.B., A.C.W., J.I.R., S.S.R., C.R.C., P.F.O., M.J., P.K., X.J., H.A., P.K.J., J.F.W., C.H., L.L., E.I., K.M., S.T., H.V., H.W., L.Z., E.T., T.S., E.T., T.L., L.L., M.K., O.T.R., V.M., M.A.I., N.S., J.W.J., N.J.W., C.L., N.G.F., J.Z., T.G., K.K., J.L., A.B., J.D., E.S., C.G., W.M., M.d.H., and M.T. reviewed the manuscript. E.H., M.I.M., E.A.S., T. J.W., D.P.K., L.A.C., R.S.V., L.F., M.P., M.J., P.K., M.C.Z., A.G.U., F.R., L.B., T.S., and M. A.I. oversee the consortium.

### Additional information

**Competing interests:** The authors declare no competing financial interests.

Xia Jiang[1,2], Paul F. O'Reilly[3], Hugues Aschard[1,4], Yi-Hsiang Hsu[5,6,7], J. Brent Richards[8], Josée Dupuis[9,10], Erik Ingelsson[11,12], David Karasik[5], Stefan Pilz[13], Diane Berry[14], Bryan Kestenbaum[15], Jusheng Zheng[16], Jianan Luan[16], Eleni Sofianopoulou[17], Elizabeth A. Streeten[18], Demetrius Albanes[19], Pamela L. Lutsey[20], Lu Yao[20], Weihong Tang[20], Michael J. Econs[21], Henri Wallaschofski[22,23], Henry Völzke[23,24], Ang Zhou[25], Chris Power[14], Mark I. McCarthy[26,27,28], Erin D. Michos[29,30], Eric Boerwinkle[31], Stephanie J. Weinstein[19], Neal D. Freedman[19], Wen-Yi Huang[32], Natasja M. Van Schoor[33], Nathalie van der Velde[34,35], Lisette C.P.G.M.de Groot[36], Anke Enneman[34], L. Adrienne Cupples[9,10], Sarah L. Booth[37], Ramachandran S. Vasan[10], Ching-Ti Liu[9], Yanhua Zhou[9], Samuli Ripatti[38], Claes Ohlsson[39], Liesbeth Vandenput[39], Mattias Lorentzon[40], Johan G. Eriksson[41,42], M. Kyla Shea[37], Denise K. Houston[43], Stephen B. Kritchevsky[43], Yongmei Liu[44], Kurt K. Lohman[45], Luigi Ferrucci[46], Munro Peacock[21], Christian Gieger[47], Marian Beekman[48], Eline Slagboom[48], Joris Deelen[48,49], Diana van Heemst[50], Marcus E. Kleber[51], Winfried März[51,52,53], Ian H. de Boer[54], Alexis C. Wood[55], Jerome I. Rotter[56], Stephen S. Rich[57,58], Cassianne Robinson-Cohen[59], Martin den Heijer[60], Marjo-Riitta Jarvelin[61,62,63,64], Alana Cavadino[14,65], Peter K. Joshi[66], James F. Wilson[66,67], Caroline Hayward[67], Lars Lind[12], Karl Michaëlsson[68], Stella Trompet[50,69], M. Carola Zillikens[60], Andre G. Uitterlinden[34,60], Fernando Rivadeneira[34,60], Linda Broer[60], Lina Zgaga[70], Harry Campbell[66,71], Evropi Theodoratou[66,71], Susan M. Farrington[71], Maria Timofeeva[71], Malcolm G. Dunlop[71], Ana M. Valdes[72,73], Emmi Tikkanen[74], Terho Lehtimäki[75,76], Leo-Pekka Lyytikäinen[75,76], Mika Kähönen[77,78], Olli T. Raitakari[79,80], Vera Mikkilä[81], M. Arfan Ikram[34], Naveed Sattar[82], J. Wouter Jukema[69,83], Nicholas J. Wareham[16], Claudia Langenberg[16], Nita G. Forouhi[16], Thomas E. Gundersen[84], Kay-Tee Khaw[17], Adam S. Butterworth[17], John Danesh[17,85], Timothy Spector[72], Thomas J. Wang[86], Elina Hyppönen[14,25], Peter Kraft[1] & Douglas P. Kiel[5,6,7]

[1]Program in Genetic Epidemiology and Statistical Genetics. Department of Epidemiology, Harvard T.H.Chan School of Public Health, 677 Huntington Avenue, Boston 02115, MA, USA. [2]Unit of Cardiovascular Epidemiology, Institute of Environmental Medicine, Karolinska Institutet, Nobels vagen 13, Stockholm 17177, Sweden. [3]Department of Social Genetic & Developmental Psychiatry, King's College London, Institute of Psychiatry, De Crespigny Park, London SE5 8AF, UK. [4]Centre de Bioinformatique, Biostatistique et Biologie Intégrative (C3BI), Institut Pasteur, Paris 75724, France. [5]Institute for Aging Research, Hebrew SeniorLife, 1200 Centre Street, Boston, MA 02131, USA. [6]Department of Medicine, Beth Israel Deaconess Medical Center and Harvard Medical School, Boston, MA 02115, USA. [7]Broad Institute of Harvard and Massachusetts Institute of Technology, Boston, MA 02142, USA. [8]Departments of Medicine, Human Genetics, Epidemiology and Biostatistics, 3755 Côte Ste-Catherine Road, Suite H-413 Montréal, Québec H3T 1E2, Canada. [9]Department of Biostatistics, Boston University School of Public Health, Crosstown Center. 801 Massachusetts Avenue

3rd Floor, Boston, MA 02118, USA. [10]Framingham Heart Study, 73 Mt. Wayte Avenue, Framingham, MA 01702, USA. [11]Department of Medicine, Division of Cardiovascular Medicine, Stanford University School of Medicine Stanford, Stanford, CA 94305, USA. [12]Department of Medical Sciences, Uppsala University, 751 85 Uppsala, Sweden. [13]Department of Internal Medicine, Division of Endocrinology and Diabetology, Medical University of Graz, Auenbruggerplatz 15, 8036 Graz, Austria. [14]Population, Policy and Practice, University College London, Great Ormond Street, Institute of Child Health, London WC1E 6BT, UK. [15]Kidney Research Institute, Division of Nephrology, 325 Ninth Avenue, Seattle, WA 98104, USA. [16]MRC Epidemiology Unit, University of Cambridge School of Clinical Medicine, Cambridge Biomedical Campus, Cambridge CB2 0QQ, UK. [17]Department of Public Health & Primary Care, University of Cambridge, Strangeways Research Laboratory, Wort's Causeway, Cambridge CB1 8RN, UK. [18]Genetics and Personalized Medicine Program, University of Maryland School of Medicine, Howard Hall Room 567, Baltimore, MD 21201, USA. [19]Metabolic Epidemiology Branch, Division of Cancer Epidemiology and Genetics, National Cancer Institute, NIH, 9609 Medical Center Drive, Bethesda, MD 20892, USA. [20]Division of Epidemiology & Community Health, School of Public Health, University of Minnesota, 1300S 2nd Street, Suite 300, Minneapolis, MN 55454, USA. [21]Department of Medicine, Indiana University, Endocrinology, 1120W Michigan Street, Indianapolis, IN 46202-5124, USA. [22]Institute of Clinical Chemistry and Laboratory Medicine, University Medicine Greifswald, 17489 Greifswald, Germany. [23]DZHK (German Centre for Cardiovascular Research), Partner Site, Greifswald, 13316 Berlin, Germany. [24]Institut für Community Medicine, SHIP/Klinisch-Epidemiologische Forschung, Universitätsmedizin Greifswald, Walther-Rathenau-Str. 48, 17475 Greifswald, Germany. [25]Centre for Population Health Research, Sansom Institute for Health Research, University of South Australia, Adelaide 5001, SA, Australia. [26]Oxford Centre for Diabetes Endocrinology and Metabolism, University of Oxford, Churchill Hospital, Old Road, Headington, Oxford OX3 7LJ, UK. [27]Wellcome Centre for Human Genetics, University of Oxford, Roosevelt Drive, Headington, Oxford OX3 7BN, UK. [28]Oxford NIHR Biomedical Research Centre, Churchill Hospital, Old Road, Headington, Oxford OX3 7LJ, UK. [29]Division of Cardiology, Ciccarone Center for the Prevention of Heart Disease, Johns Hopkins School of Medicine, Baltimore, MD 21287, USA. [30]Department of Epidemiology, Johns Hopkins Bloomberg School of Public Health, Baltimore, MD 21205, USA. [31]Human Genetics Center, University of Texas Health Science Center at Houston, Houston, TX 77030, USA. [32]Occupational and Environmental Epidemiology Branch, Division of Cancer Epidemiology and Genetics, National Cancer Institute, NIH, 9609 Medical Center Drive, Bethesda, MD 20892, USA. [33]Department of Epidemiology and Biostatistics, Amsterdam Public Health Research Institute, VU University Medical Center, De Boelelaan 1089a, 1081 HV Amsterdam, The Netherlands. [34]Erasmus MC Department of Epidemiology, Postbus 2040, 3000CA Rotterdam, The Netherlands. [35]AMC, Internal Medicine, Geriatrics Department, PO Box 227001100 DE Amsterdam, The Netherlands. [36]Department of Human Nutrition, Wageningen University, PO-box 176700 AA Wageningen, The Netherlands. [37]Vitamin K Laboratory, Jean Mayer USDA Human Nutrition Research Center on Aging, Tufts University, 711 Washington Street, Boston, MA 02111, USA. [38]Statistical and Translational Genetics, University of Helsinki, Tukholmankatu 8, Building, Biomedicum, Helsinki 2U, Finland. [39]Department of Internal Medicine and Clinical Nutrition, University of Gothenburg, Vita Stråket 11, Gothenburg 41345, Sweden. [40]Department of Geriatric Medicine, University of Gothenburg and Sahlgrenska University Hospital, Mölndal 43180, Sweden. [41]Department of General Practice and Primary Health Care, University of Helsinki and Helsinki University Hospital, University of Helsinki, P.O. Box 20, Tukholmankatu 8 B 00014, Finland. [42]Folkhälsan Research Center, University of Helsinki, Helsinki PO Box 2000014, Finland. [43]Sticht Center for Healthy Aging and Alzheimer's Prevention, Wake Forest School of Medicine, Medical Center Boulevard, Winston-Salem, NC 27157, USA. [44]Department of Epidemiology and Prevention, Division of Public Health Sciences, Wake Forest School of Medicine, Medical Center Blvd, Winston-Salem, NC 27157, USA. [45]Department of Biostatistical Sciences, Division of Public Health Sciences, Wake Forest School of Medicine, Medical Center Blvd, Winston-Salem, NC 27157, USA. [46]Longitudinal Studies Section, Intramural Research Program of the National Institute on Aging, NIH, Baltimore, MD 21225, USA. [47]German Research Center for Environmental Health, Molecular Epidemiology, AME, Ingolstädter Landstr 1, D-85764 Neuherberg, Germany. [48]Molecular Epidemiology, Leiden University Medical Center, Einthovenweg 20, 2333 ZC Leiden, The Netherlands. [49]Max Planck Institute for Biology of Ageing, Joseph-Stelzmann-Str. 9b, D-50931 Köln (Cologne), Germany. [50]Gerontology and Geriatrics, Leiden University Medical Center, Albinusdreef 2, 2333 ZA Leiden, The Netherlands. [51]Vth Department of Medicine (Nephrology, Hypertensiology, Rheumatology, Endocrinology, Diabetology), Medical Faculty Mannheim, University of Heidelberg, Theodor-Kutzer-Ufer1, 68167 Mannheim, Germany. [52]Clinical Institute of Medical and Chemical Laboratory Diagnostics, Medical University of Graz, Auenbruggerplatz 15, 8036 Graz, Graz, Austria. [53]SYNLAB Holding Deutschland GmbH, Gubener Straße 39, 86156 Augsburg, Germany. [54]Division of Nephrology and Kidney Research Institute, University of Washington, 325 ninth Avenue, Washington, DC 98104, USA. [55]USDA/ARS Children's Nutrition Research Center, 1100 Bates Avenue, Houston, TX 77071, USA. [56]Institute for Translational Genomics and Population Sciences, Los Angeles Biomedical Research Institute and Department of Pediatrics, Harbor-UCLA Medical Center, Torrance, CA 90502, USA. [57]Department of Public Health Sciences, University of Virginia, Charlottesville, VA 22908, USA. [58]Center for Public Health Genomics, University of Virginia, Charlottesville, VA 22908, USA. [59]Division of Nephrology, Department of Medicine, Vanderbilt University Medical Center, 1161 21st Ave S., Nashville, TN 37232, USA. [60]Erasmus MC Department of Internal Medicine, Postbus 20403000CA Rotterdam, The Netherlands. [61]Epidemiology and Biostatistics School of Public Health, Imperial College London, 156 Norfolk Place, St. Mary's Campus, London UK W2 1PG, UK. [62]Center for Life Course Health Research, Faculty of Medicine, University of Oulu, 90014 Oulu, Finland. [63]Biocenter Oulu, University of Oulu, P.O. Box 5000, Aapistie 5A FI-90014, Finland. [64]Unit of Primary Care, Oulu University Hospital, Kajaanintie 50P.O. Box 20FI-90220 Oulu, 90029 OYS, Finland. [65]Centre for Environmental and Preventive Medicine, Wolfson Institute of Preventive Medicine, Barts and The London School of Medicine and Dentistry, Queen Mary University of London, Charterhouse Square, London EC1M 6BQ, UK. [66]Centre for Global Health Research, Usher Institute for Population Health Sciences and Informatics, University of Edinburgh, Teviot Place, Edinburgh EH8 9AG, UK. [67]MRC Human Genetics Unit, MRC Institute of Genetics & Molecular Medicine, the University of Edinburgh, Western General Hospital, Edinburgh EH4 2XU, UK. [68]Department of Surgical Sciences, Uppsala University, Dag Hammarskjöldsv 14 B, Uppsala Science Park, 751 85 Uppsala, Sweden. [69]Department of Cardiology, Leiden University Medical Center, Albinusdreef 2, 2333 ZA Leiden, Netherlands. [70]Department of Public Health and Primary Care, Institute of Population Health, Trinity College Dublin, University of Dublin, Dublin 24 D02 PN40, Ireland. [71]Institute of Genetics and Molecular Medicine, University of Edinburgh, Western General Hospital, Edinburgh EH4 2XU, UK. [72]The Department of Twin Research & Genetic Epidemiology, King's College London, St Thomas' Campus, Westminster Bridge Road, London SE1 7EH, UK. [73]School of Medicine, University of Nottingham, City Hospital, Hucknall Rd, Nottingham NG5 1PB, UK. [74]FIMM-Institute for Molecular Medicine Finland, University of Helsinki, HelsinkiP.O. Box 20FI-00014, Finland. [75]Department of Clinical Chemistry, Fimlab Laboratories, Tampere 33520, Finland. [76]Department of Clinical Chemistry, Finnish Cardiovascular Research Center Tampere, Faculty of Medicine and Life Sciences, University of Tampere, Tampere 33014, Finland. [77]Department of Clinical Physiology, Tampere University Hospital, Tampere 33521, Finland. [78]Department of Clinical Physiology, Finnish Cardiovascular Research Center Tampere, Faculty of Medicine and Life Sciences, University of Tampere, Tampere 33014, Finland. [79]Department of Clinical Physiology and Nuclear Medicine, Turku University Hospital, Turku 20521, Finland. [80]Research Centre of Applied and Preventive Cardiovascular Medicine, University of Turku, Turku 20014, Finland. [81]Science Adviser at Academy of Finland, Hakaniemenranta 6, PO Box 131, FI-00531 Helsinki, Finland. [82]BHF Glasgow Cardiovascular Research Centre, Faculty of Medicine, University Avenue, Glasgow G12 8QQ, UK. [83]Einthoven Laboratory for Experimental Vascular Medicine, Leiden University Medical Center, Albinusdreef 2, 2333 ZA Leiden, The Netherlands. [84]Vitas AS, Gaustadaleen 21, N-0349 Oslo, Norway. [85]Wellcome Trust Sanger Institute, Wellcome Genome Campus, Hinxton, Cambridge CB10

1SA, UK. [86]Division of Cardiovascular Medicine, Vanderbilt Heart and Vascular Institute, 2220 Pierce Avenue 383 Preston Research Building, Nashville, TN 37232-6300, USA. Xia Jiang, Paul F. O'Reilly, Hugues Aschard and Yi-Hsiang Hsu contributed equally to this work. Thomas J. Wang, Elina Hyppönen, Peter Kraft and Douglas P. Kiel jointly supervised this work.

