## [Peer Review File · Nature Communications]

Reviewers' comments:

Reviewer #1 (Remarks to the Author):

This study expanded their previous GWAS by adding 21 additional cohorts, and performed a single stage discovery meta-analysis on a total of up to 79,366 individuals. The results from a fixed-effects inverse variance weighted meta-analysis across the contributing cohorts confirmed their four previously reported loci and revealed two novel loci at AMDHD1 (rs10745742) and SEC23A (rs8018720) that achieved genome wide significance

Major comments

1. Sometimes, PCA may not be sufficient to control for population stratification, especially for large datasets. What is the genomic inflation factor for the QQ plot of Figure1? The genomic inflation factor for the QQ plot of Supplementary Figure 2a, which controlled for vitamin D intake, was given as 1.03. Comparison of the QQ plot of Figure1 and Supplementary Figure 2a suggests that the QQ plot of Figure1 may have a much higher genomic inflation factor that may not be acceptable. QQ plot and genomic inflation factor should also be given for each of the 31 participating cohorts in supplementary figures, as even though the combined statistics from meta-analysis are not unacceptably inflated, it does not necessarily mean that population stratification was well under controlled for each participating cohort.
2. What's the p values for rs10745742 and rs8018720 from the marginal genetic effect tests after controlling for vitamin D intake in the same sub-samples? As no significant interactions were detected between any SNPs and dietary vitamin D intake, the results in table 1 for the interaction analysis should be replaced with results from the marginal genetic effect test for each SNP controlling for vitamin D intake.
3. It would be more convincing if replicated associations can be show for the two novel loci in independent datasets
4. In the methods section, it says that all SNPs with imputation info score <0.3 were removed, whereas in the results section, it says that SNPs with imputation info score >0.8 were retained.

Minor comments

1. r^2 equals 1.0 for SNP pairs rs3755967 and rs2282679, and rs12785878 and rs4944062, indicating that SNPs of each pair are in perfect LD.
2. rs3755967 in table1 and rs2282679 in supplementary table 3 have opposite effect directions.

Reviewer #2 (Remarks to the Author):

This is a large collaborative GWAS on the genetic underpinnings of vitamin D (VD) variation, using close to 80,000 individuals of European descent. With a much larger sample size than previous GWAS, 2 new loci are identified that explain roughly 1/3 of gene estimated genetic contribution to VD variation (about 10%). Signal-summary tests point towards immune cells. There is no interaction demonstrated with dietary patterns.

The manuscript is clearly written and there are experts in the field of GWAS among the first and last authors. I find this an interesting read, but have a few comments:

MAJOR

- 1) The argument on heritability is confused: The abstract states 8% due to GWAS SNPs (common SNPs - not all readers will recognize this), but overall heritability is much larger as the text states. Consequently I find the conclusion that VD heritability is due to few SNPs misguided. How does this compare to other complex traits? The partitioning experiments are nice, but may also be compared to other traits to be more meaningful.

- 2) There is no replication of the findings.
- 3) The Manhattan and QQ plot might show new and previous findings separately.
- 4) There is surprisingly little information on the phenotype per study and the comparability between different measurement methods. I would wish to see some metrics of distribution.
- 5) I cannot find any information on genomic control and the stages at which it was applied. Many of the participating studies are case-control, does this influence the results?
- 6) The number of significant SNPs is low - how does this influence and potentially bias downstream analyses?
- 7) The authors might point out more clearly that statistical power is probably quite low in the interaction tests (how low?), increasing the probability of a type 2 error.
- 8) Where will the summary statistics be posted?

Other

- This reviewer perceives the abstract as overenthusiastic on the sample size. Yes, the sample size is much increased, but remains perhaps "small", given the important multiple-testing burden that the authors face given the total number of tests. A slight change in the abstract and text might be useful.
- Figure 4 may go into the supplement.

Reviewer #3 (Remarks to the Author):

The manuscript describes a genome-wide association study of circulating vitamin D levels in 80,000 individuals. This is a substantial increase in sample size from 16,000 individuals in the previous iteration of the SUNLIGHT consortium analysis. Despite this increase in sample size only 2 additional genome-wide significant loci are identified. Based on this and the fact that a substantial amount of the chip-based heritability is explained by GWAS significant loci, the authors major conclusion appears to be that variance in circulating Vitamin D levels is explained by only a small number of loci.

This is a potentially interesting manuscript and the new association signals seem robust. However, I am less sure about the other conclusions from the manuscript.

1. The conclusion that only a small number of loci explain most of the variance in circulating Vitamin D levels from common variants doesn't seem all that interesting or important a conclusion and I don't think it is strongly supported by the data. There are a couple of very strong common variants signals for Vitamin D levels and these dominate the common variant associations so I think the conclusion is somewhat obvious. This conclusion also doesn't include low frequency or rare variants given the authors have used a 5% MAF cut-off in their analyses. They also rightly point to the large number of potential environment covariates in and between studies that might affect the conclusions.
2. Why was only HapMap imputation used? How would the conclusions have changed if much bigger panels (like the 1000 genomes or HRC) had been used as the reference panel instead? Why was a $MAF < 0.05$ cut-off used. It seems slightly outdated to only use HapMap imputation and not to assess lower frequency variants.
3. The authors conclusion that Vitamin D levels are causal to autoimmune diseases is also questionable based on the analyses presented in the manuscript. None of the genetic correlation analyses supports this and, in my view, the epigenetic and tissue enrichment analyses provides almost no evidence for this conclusion. There is perhaps, visually, some similarity to autoimmune diseases in the pattern of Vitamin D enrichment, but it is not entirely clear from Figure 4 and statistics are needed to support this. Even if there was a statistically robust similarity with autoimmune diseases I'm not

sure what strong conclusions can be drawn from this anyway. The authors should include Mendelian Randomisation analyses to support their conclusion.

Reviewer #1 (Remarks to the Author):

This study expanded their previous GWAS by adding 21 additional cohorts, and performed a single stage discovery meta-analysis on a total of up to 79,366 individuals. The results from a fixed-effects inverse variance weighted meta-analysis across the contributing cohorts confirmed their four previously reported loci and revealed two novel loci at AMDHD1 (rs10745742) and SEC23A (rs8018720) that achieved genome wide significance

Major comments

1. Sometimes, PCA may not be sufficient to control for population stratification, especially for large datasets. What is the genomic inflation factor for the QQ plot of Figure1? The genomic inflation factor for the QQ plot of Supplementary Figure 2a, which controlled for vitamin D intake, was given as 1.03. Comparison of the QQ plot of Figure1 and Supplementary Figure2a suggests that the QQ plot of Figure1 may have a much higher genomic inflation factor that may not be acceptable. QQ plot and genomic inflation factor should also be given for each of the 31 participating cohorts in supplementary figures, as even though the combined statistics from meta-analysis are not unacceptably inflated, it does not necessary mean that population stratification was well under controlled for each participating cohort.

Answer: We have now labeled in Figure 1 that the genomic inflation factor for the QQ plot using all samples was 0.99. It did not exceed the genomic inflation factors in the analysis where we controlled dietary intake of vitamin D, indicating that our GWAS results were not inflated by population stratification or cryptic relatedness. Furthermore, as requested by the reviewer, we performed QQ plots for all contributing cohorts and none of the individual cohort QQ plots demonstrated any evidence for inflation.

We added a sentence stating these results. It reads, *“To assess and control for population stratification, we examined QQ-plots and genomic control inflation factors for each contributing cohort prior to meta-analysis. We did not observe evidence for widespread inflation (median $\lambda_{GC}=0.92$; only 1/31 samples with $\lambda_{GC}>1.01$), indicating that our GWAS results were not inflated by population stratification or cryptic relatedness (Supplementary Figure 1)”*

QQ-plots for each individual cohort are shown below with lambda values presented on top of each plot. We performed analysis in cases and controls separately when the participating cohort is in a case-control design. We did not observe any signs of inflation.

2. What's the p values for rs10745742 and rs8018720 from the marginal genetic effect tests after controlling for vitamin D intake in the same sub-samples? As no significant interactions were detected between any SNPs and dietary vitamin D intake, the results in table 1 for the interaction analysis should be replaced with results from the marginal genetic effect test for each SNP controlling for vitamin D intake.

Answer: We assume by saying “the same sub-samples” the reviewer is referring to the sub-sample used in the interaction analysis. The *P*-values for marginal association from the vitamin D intake adjusted model (in the sub-sample used in the interaction effect analysis) are presented in Supplementary Table 1. Note that this table only includes the most significant SNP per locus. For rs10745742 the *P*-value is 5.7×10^{-9} , as shown in Supplementary Table 3. The *P*-value for rs8018720 is 8.94×10^{-5} . We have now added the latter SNP to Supplementary Table 3.

We decided to present results from the interaction analysis and main genetic effect as we believe the readers might be interested by this specific analysis – i.e. whether SNPs found in the meta-analysis display interaction effects.

Marginal genetic effect tests in the same sub-sample are mostly relevant for the power comparison with joint tests, which is not the primary purpose of our study, and was therefore presented in supplementary data. Indeed, because of a dramatically smaller sample size in the sub-sample used for interaction analysis, as compared to the full meta-analysis ($N=41,981$ and $N=79,366$, respectively), we do not expect any new signal to be identified by either the marginal or joint tests in these analyses.

3. It would be more convincing if replicated associations can be show for the two novel loci in independent datasets

Answer: Following the reviewer’s suggestion, we replicated the novel loci in two independent datasets: the European Prospective Investigation into Cancer and Nutrition (EPIC) study with 40,562 individuals

collected from two nested case-control studies (EPIC-InterAct, EPIC-CVD) and one cohort-wide study (EPIC-Norfolk) (Supplementary Methods); and a cohort of 2,195 individuals (all controls) additionally collected as part of SOCCS that were not included in our discovery stage. Genotype data for the two SNPs were available for all individuals. As for the phenotype, EPIC individuals were assayed for plasma 25-hydroxyvitamin D₃ and SOCCS individuals were assayed for total 25-hydroxyvitamin D. We performed association analysis in a similar manner, adjusted for age, sex, time of sample collection, and study center where relevant. We declared a successful replication if P -value < 0.05 in the replication samples, and P -value $< 5 \times 10^{-8}$ in the pooled analysis. The association at both novel loci were confirmed in the two independent *in-silico* replication cohorts (EPIC: $P=1.21 \times 10^{-8}$ at rs10745742, $P=5.24 \times 10^{-4}$ at rs8018720; SOCCS: $P=0.03$ at rs10745742, $P=0.04$ at rs8018720) with consistent direction of effect. When analyzing the two replication datasets together with the discovery dataset, the pooled P -values became more significant ($P_{pooled}=2.10 \times 10^{-20}$ at rs10745742, $P_{pooled}=1.11 \times 10^{-11}$ at rs8018720). We have added these results to our manuscript.

Table 1. Results from the Meta-analysis, and the SNP-by-dietary vitamin D intake interaction analysis.

Gene	SNP	Chromosome: Position	Effect/ Reference Allele	Allele Frequency	Meta-analysis		
					Effect (Beta)	Standard Error	P-value
First stage discovery meta-GWAS (N=79,366)							
GC	rs3755967	4:72828262	T/C	0.28	-0.089	0.0023	4.74E-343
	rs2282679*	4:72827247	T/G	0.28	0.089	0.0024	1.65E-314
NADSYN1/ DHCR7	rs12785878 rs4944062*	11:70845097 11:70864942	T/G T/G	0.75 0.75	0.036 0.036	0.0022 0.0023	3.80E-62 1.48E-53
CYP2R1	rs10741657	11:14871454	A/G	0.40	0.031	0.0022	2.05E-46
CYP24A1	rs17216707	20:52165769	T/C	0.79	0.026	0.0027	8.14E-23
AMDHD1	rs10745742	12:94882660	T/C	0.40	0.017	0.0022	1.88E-14
SEC23A	rs8018720	14:38625936	C/G	0.82	-0.017	0.0029	4.72E-09
Replication dataset 1: samples collected by EPIC (N=40,562)							
AMDHD1	rs10745742	12:94882660	T/C	0.41	0.041	0.0071	1.21E-08
SEC23A	rs8018720	14:38625936	C/G	0.83	-0.032	0.0093	5.24E-04
Replication dataset 2: additional control samples collected by SOCCS (N=2,195)							
AMDHD1	rs10745742	12:94882660	T/C	0.37	0.045	0.021	0.03
SEC23A	rs8018720	14:38625936	C/G	0.81	-0.051	0.026	0.04
Pooled analysis (discovery meta-GWAS + replication 1 + replication 2) (N=122,123)							
AMDHD1	rs10745742	12:94882660	T/C	0.39	0.019	0.0020	2.10E-20
SEC23A	rs8018720	14:38625936	C/G	0.82	-0.019	0.0027	1.11E-11

4. In the methods section, it says that all SNPs with imputation info score < 0.3 were removed, whereas in the results section, it says that SNPs with imputation info score > 0.8 were retained.

Answer: Thank you for pointing out the two different thresholds that were used. We indeed made slightly different decisions on the quality control of our two main analyses. For the first stage discovery meta-analysis, we used SNPs of high imputation quality (retained SNPs with imputation info score > 0.8). For the gene-by-diet interaction analysis, where the sample size was greatly decreased (reduced to only half of the original samples), we applied less stringent inclusion criteria (retained SNPs with imputation info score > 0.3) to include more candidates. Results from the two analyses were highly concordant. We have now used a more stringent threshold for the gene-by-diet interaction analysis and we did not see substantial changes in the results. We have moved the paragraph of quality control procedures from the results section to the methods section, as well as making some clarifications.

Minor comments

1. r^2 equals 1.0 for SNP pairs rs3755967 and rs2282679, and rs12785878 and rs4944062, indicating that SNPs of each pair are in perfect LD.

Answer: We have clarified this in our results section. It reads: “*The marginal genetic effect analyses confirmed existing association signals at GC (lead SNP rs2282679, in complete linkage disequilibrium with the lead SNP rs3755967 identified from the meta-GWAS using all individuals), CYP2R1, NADSYN1/DHCR7 (lead SNP rs4944062, in complete linkage disequilibrium with the lead SNP rs12785878 identified from the meta-GWAS using all individuals) and CYP24A1, as well as the novel association at gene AMDHD1 ($P=5.7\times 10^{-9}$, Supplementary Table 3)*”.

2. rs3755967 in table1 and rs2282679 in supplementary table 3 have opposite effect directions.

Answer: We thank the reviewer for noting this potentially confusing result. The opposite effect is simply due to the arbitrary choice of coded allele we made. Indeed, the two SNPs are in complete LD. In the original coding the correlation between the two SNPs equals to -1, and the estimated effect is therefore in opposite direction. Flipping the coded/reference allele for either SNP, we obtain, as expected, a concordant effect.

Reviewer #2 (Remarks to the Author):

This is a large collaborative GWAS on the genetic underpinnings of vitamin D (VD) variation, using close to 80,000 individuals of European descent. With a much larger sample size than previous GWAS, 2 new loci are identified that explain roughly 1/3 of gene estimated genetic contribution to VD variation (about 10%). Signal-summary tests point towards immune cells. There is no interaction demonstrated with dietary patterns. The manuscript is clearly written and there are experts in the field of GWAS among the first and last authors. I find this an interesting read, but have a few comments:

MAJOR

1. The argument on heritability is confused: The abstract states 8% due to GWAS SNPs (common SNPs - not all readers will recognize this), but overall heritability is much larger as the text states. Consequently, I find the conclusion that VD heritability is due to few SNPs misguided. How does this compare to other complex traits? The partitioning experiments are nice, but may also be compared to other traits to be more meaningful.

Answer: The overall heritability estimated from classical twin studies taking into account all genetic effects (common variants, rare variants, gene-gene interactions, and more) is indeed much higher than the SNP-heritability calculated by us using all common SNPs on the GWAS chip. We have highlighted this difference in our discussion, it reads “*Heritability estimates obtained using common SNPs in large GWAS have typically been found to be approximately half those from classical twin studies^{9,10}, but our estimate of 7.54%, calculated using common genome-wide SNPs, is far lower than reported heritability from twin and family based studies. In addition to potentially inflated estimates from twin studies, the difference may reflect the proportion of heritability explained by rare SNPs or structural variants that were not included in our data, and potential gene-gene interactions that remain to be identified.*”

We collected estimates of heritability for a few other traits from well-conducted large-scale twin studies and GWA studies. The following plot demonstrates that although the heritability varies across different traits, a comparable pattern is observed. For most of the traits, SNP-heritability accounts for less than half of the heritability estimated from twin studies (e.g., vitamin D, rheumatoid arthritis, schizophrenia, kidney cancer, bladder cancer). Of the total SNP-heritability estimated using SNPs on the GWAS chip, a good proportion could be explained by known GWAS-identified hits (for height, 11%; testis cancer: 33%; prostate cancer: 24%; rheumatoid arthritis: 30%). (raw data extracted from: Sampson *et al. JNCI (2015) 107(21): djv279*; Mucci *et al. JAMA 2016;315(1):68-76*)

Twin heritability vs. SNP-heritability

We have further clarified our conclusion as “*The overall estimate of heritability of 25-hydroxyvitamin D serum concentrations attributable to GWAS common SNPs was 7.5%, with statistically significant loci explaining 38% of this total (2.8% out of 7.5%).*” “*Larger studies of this phenotype are required to identify additional common SNPs, as well as to explore the role of rare or structural variants, and gene-gene interactions in the heritability of circulating 25-hydroxyvitamin D levels.*”

In Figure 4, we partitioned the SNP-heritability of circulating 25-hydroxyvitamin D as well as 37 additional traits on 220 cell-type-specific annotations. We compared the pattern of enrichment among those 38 phenotypes by clustering both traits and annotations in heat-maps. As shown in Figure 4, the phenotypes belonging to the same disease category exhibited comparable patterns of enrichment. For example, psychiatric disorders such as schizophrenia, anorexia, bipolar, depressive symptoms were clustered to the same group, along with smoking and age at menarche, consistent with already established knowledge. We found that circulating 25-hydroxyvitamin D exhibited comparable patterns of enrichment with autoimmune inflammatory diseases, where multiple immune cells were enriched.

2. There is no replication of the findings.

Answer: As the reviewer suggested, we have replicated our novel loci in two independent datasets where both genotypes and plasma 25-hydroxyvitamin D levels were measured (EPIC: N=40,562; SOCCS: N=2,195). The association at both novel loci were confirmed in the two independent *in-silico* replication cohorts (EPIC: $P=1.21 \times 10^{-8}$ at rs10745742, $P=5.24 \times 10^{-4}$ at rs8018720; SOCCS: $P=0.03$ at rs10745742, $P=0.04$ at rs8018720) with consistent direction of effect. We have added these results to the manuscript. For a detailed description please also read the answers responding to question #3 reviewer #1.

3. The Manhattan and QQ plot might show new and previous findings separately.

Answer: As suggested by both reviewer #1 and #2, we have made changes to the Manhattan plot and QQ plot. In the current Manhattan plot, known loci were color coded as red, and novel loci were color coded as green. We also added lambda values to the QQ plot.

A) Manhattan Plot

B) Q-Q plot of GWAS p-values

4. There is surprisingly little information on the phenotype per study and the comparability between different measurement methods. I would wish to see some metrics of distribution.

Answer: We have plotted the mean (SD) of circulating 25-hydroxyvitamin D levels for the 10 GWAS cohorts collected by the year 2010 (where we have already existing data). We found comparable distributions for the concentration of circulating 25-hydroxyvitamin D across the 10 cohorts irrespective of assay platforms or measurement methods.

These previous results demonstrated that the varying assays used by different cohorts showed similar distribution of 25-hydroxyvitamin D concentrations.

Figure1. Mean (SD) of circulating 25-hydroxyvitaminD levels across the 10 GWAS cohorts collected by year 2010

5. I cannot find any information on genomic control and the stages at which it was applied. Many of the participating studies are case-control, does this influence the results?

Answer: Genomic control was applied to each of the 31 individual studies before meta-analysis. We did not apply a second level of genomic control on our meta-GWAS, as we did not observe genomic inflation (see QQ-plots and response to Reviewer #1 question #1, the meta-analysis lambda for overall samples was 0.99). We used GWAS data from multiple case-control studies including type II diabetes, coronary heart disease, colorectal cancer, and more. There might be a potential concern that including case subjects might introduce bias. However, our group has shown through theoretical (Monsees *et al. Genet Epidemiol*, 33:717-728, 2009) and empirical (Lindstrom *et al. Nat Genet*, 43:185-7, 2011) approaches that including case subjects does not affect the validity of top findings. Furthermore, we performed the GWAS separately in cases and controls before meta-analyzing the results.

6. The number of significant SNPs is low - how does this influence and potentially bias downstream analyses?

Answer: The fact that only a small number of loci achieve genome-wide significance does not indicate that the point estimates of heritability, functional enrichment, or cross heritability are biased. A key advantage of LD score regression is that it leverages the aggregate signal (non-zero test-statistic non-centrality parameters) across many non-null markers (while adjusting for potential bias due to population stratification) to estimate the overall contribution of common SNPs to variance in a trait; these calculations are not limited to the number of genome-wide significant variants. Moreover, the point estimates of heritability are consistent regardless of the shape of the distribution of genetic effects — so, for example, heritability estimates were unbiased in simulations where only a small fraction of variants were causal (Bulik-Sullivan *et al. Nat Genet*. 2015 Mar; 47(3):291-5) or where a subset of variants had larger effects than others (Finucane *et al. Nature Genetics* 2015 Nov; 47(11):1228-1235). The variance of the heritability estimate, on the other hand, is affected by the distribution of the genetic effects. In situations where a small number of variants disproportionately contribute to heritability the variance is increased —yet despite this inefficiency we observed a statistically significant common-SNP heritability ($h_g^2=7.54\%$, $P\text{-value} = 6\times 10^{-5}$).

7. The authors might point out more clearly that statistical power is probably quite low in the interaction tests (how low?), increasing the probability of a type 2 error.

Answer: Indeed, power for interaction effect is known to be much smaller than for marginal effects. We agree with the reviewer that many SNPs involved in gene-environment interactions remain to be discovered, just as many SNPs with marginal effects remain to be discovered. That said, we did have power to detect gene-environment interaction effects even smaller than the observed marginal effects. In the case where a SNP has no marginal effect on circulating vitamin D concentrations (and so could not have been discovered via the marginal GWAS), we had 80% power to detect an interaction that explained 0.07% of the total variance in 25-hydroxyvitamin D concentrations (Power calculated using Quanto, <http://biostats.usc.edu/Quanto.html>). Thus, while smaller gene-diet interaction effects remain to be discovered, our results provide some evidence against large interactions between common SNPs and dietary vitamin D intake. We have added a paragraph in methods and discussion to illustrate power of our gene-environment interaction analyses.

8. Where will the summary statistics be posted?

Answer: We will follow the data policies of *Nature Communications*, and deposit the summary statistics data in a persistent repository where they can be freely and enduringly accessed. We will post our data under dbGap (CHARGE site) for people to download <https://www.ncbi.nlm.nih.gov/gap>. We will share our summary statistics data via LDhub <http://ldsc.broadinstitute.org/gwashare/>.

Other

1. This reviewer perceives the abstract as overenthusiastic on the sample size. Yes, the sample size is much increased, but remains perhaps "small", given the important multiple-testing burden that the authors face given the total number of tests. A slight change in the abstract and text might be useful.

Answer: As requested we have now amended the text slightly. It reads, "*In this study, we expand our previous SUNLIGHT Consortium GWAS discovery sample size five-fold from 16,125 to 79,366. In contrast to other GWAS with sample size increases of similar magnitude, this larger GWAS yielded only two additional loci harboring genome-wide significant variants ($P=4.7 \times 10^{-9}$ at rs8018720 in SEC23A, and $P=1.9 \times 10^{-14}$ at rs10745742 in AMDHD1).*"

2. Figure 4 may go into the supplement.

Answer: Figure 4 partitioned the SNP-heritability of circulating 25-hydroxyvitamin D as well as 37 additional phenotypes on 220 cell-type-specific annotations, and compared the patterns of enrichment, which, might be of interest to some of our readers. We defer to the editor as to whether Figure 4 should be moved to supplemental figures. Because so many supplemental figures are not examined by readers, we feel these results will be interesting to readers of this manuscript and prefer to leave them in the main paper.

Reviewer #3 (Remarks to the Author):

The manuscript describes a genome-wide association study of circulating vitamin D levels in 80,000 individuals. This is a substantial increase in sample size from 16,000 individuals in the previous iteration of the SUNLIGHT consortium analysis. Despite this increase in sample size only 2 additional genome-wide significant loci are identified. Based on this and the fact that a substantial amount of the chip-based heritability is explained by GWAS significant loci, the authors major conclusion appears to be that variance in circulating Vitamin D levels is explained by only a small number of loci.

This is a potentially interesting manuscript and the new association signals seem robust. However, I am less sure about the other conclusions from the manuscript.

1. The conclusion that only a small number of loci explain most of the variance in circulating Vitamin D levels from common variants doesn't seem all that interesting or important a conclusion and I don't think it is strongly supported by the data. There are a couple of very strong common variants signals for

Vitamin D levels and these dominate the common variant associations so I think the conclusion is somewhat obvious. This conclusion also doesn't include low frequency or rare variants given the authors have used a 5% MAF cut-off in their analyses. They also rightly point to the large number of potential environment covariates in and between studies that might affect the conclusions.

Answer: As pointed out by the reviewer, we are aware of the influence of rare variants, gene-gene interactions, and potential environmental factors on our conclusion. It is beyond the scope of our current manuscript to study rare variants, notably as related efforts have recently been published (see AJHG, 2017 Aug 3; 101(2):227-238). We have further modified our conclusion as “*The overall estimate of heritability of 25-hydroxyvitamin D serum concentrations attributable to GWAS common SNPs was 7.5%, with statistically significant loci explaining 38% of this total (2.8% out of 7.5%).*” “*Larger studies of this phenotype are required to identify additional common SNPs, as well as to explore the role of rare or structural variants, and gene-gene interactions in the heritability of circulating 25-hydroxyvitamin D levels.*”

2. Why was only HapMap imputation used? How would the conclusions have changed if much bigger panels (like the 1000 genomes or HRC) had been used as the reference panel instead? Why was a MAF<0.05 cut-off used. It seems slightly outdated to only use HapMap imputation and not to assess lower frequency variants.

Answer: Since in this study we expanded the sample size from a previous project for which most cohorts performed their analysis using data imputed to the HapMap2 reference panel, it would have been logistically challenging to request these cohorts to re-impute their results using another reference panel given limited available resources.

As pointed out by the reviewer and answered in the previous question, we are aware of the influence from rare variants, gene-gene interactions, and potential further environmental factors on our conclusion, and investigation of these influences is beyond the scope of our study. We have added a paragraph in the discussion, which reads, “*In addition to potentially inflated estimates from twin studies, the difference may reflect the proportion of heritability explained by rare SNPs or structural variants that were not included in our data, and the potential gene-gene and gene-environment interactions that remain to be identified.There also may be low frequency variants with larger effects that were not studied here.....*”.

3. The authors conclusion that Vitamin D levels are causal to autoimmune diseases is also questionable based on the analyses presented in the manuscript. None of the genetic correlation analyses supports this and, in my view, the epigenetic and tissue enrichment analyses provide almost no evidence for this conclusion. There is perhaps, visually, some similarity to autoimmune diseases in the pattern of Vitamin D enrichment, but it is not entirely clear from Figure 4 and statistics are needed to support this. Even if there was a statistically robust similarity with autoimmune diseases I'm not sure what strong conclusions can be drawn from this anyway. The authors should include Mendelian Randomization analyses to support their conclusion.

Answer: We agree with the reviewer that our analyses alone do not prove or disprove a causal relationship between vitamin D levels and autoimmune disease. To avoid ambiguity, we have further modified the discussion: “*These findings are in line with previous Mendelian Randomization studies which found a putative causal association between vitamin D and autoimmune diseases such as multiple sclerosis^{1,2} and type 1 diabetes²⁴.*”

Of note, we have included a directional genetic correlation analysis using the methods published by Pickrell *et al.* (*Nature Genetics* 2016 Jul;48(7):709-17), which adopts a similar intuition as the Mendelian Randomization approach suggested by the reviewer. In this method, if a trait X influences trait Y, then SNPs influencing X should also influence Y, and the SNP-specific effect sizes for the two traits should be correlated. Further, since Y does not influence X, but could be influenced by mechanisms independent of X, genetic variants that influence Y do not necessarily influence X. Based

on this intuition, the method proposes two “causal” models and two “non-causal” models, and calculates the relative likelihood ratio of the best non-causal model compared to the best causal model. However, as discussed on page 4, we did not find apparent putative causal relationships between vitamin D and other traits, probably due to having only six SNPs associated with 25-hydroxyvitamin D.

REVIEWERS' COMMENTS:

Reviewer #1 (Remarks to the Author):

All my comments have been adequately addressed. The only thing I want to point out is that the GWAS statistics of most of the constituent cohorts were quite deflated with $\lambda < 0.93$ due to over-correction of test statistics, which will decrease the power to detect new associations.

Reviewer #2 (Remarks to the Author):

Excellent rebuttal. No further comments.

Reviewer #3 (Remarks to the Author):

The authors have adequately addressed my comments

REVIEWERS' COMMENTS:

Reviewer #1 (Remarks to the Author):

All my comments have been adequately addressed. The only thing I want to point out is that the GWAS statistics of most of the constituent cohorts were quite deflated with $\lambda < 0.93$ due to over-correction of test statistics, which will decrease the power to detect new associations.

Answer: We thank the reviewer for pointing this out. We have added a sentence in the results section, it reads: "Despite the slightly deflated λ_{GC} observed in some of the constituent cohorts most probably due to over-correction of test statistics, our λ_{GC} of 0.99 in all samples indicated appropriate control for population stratifications and confounders."

Reviewer #2 (Remarks to the Author): Excellent rebuttal. No further comments.

Answer: We thank the reviewer for this positive comment.

Reviewer #3 (Remarks to the Author): The authors have adequately addressed my comments

Answer: We thank the reviewer for this positive comment.